# Identification of structural and regulatory cell-shape determinants in *Haloferax volcanii*

Heather Schiller [1], Yirui Hong [1], Joshua Kouassi[1], Theopi Rados [2], Jasmin Kwak[2], Anthony DiLucido [1], Daniel Safer [3], Anita Marchfelder [4], Friedhelm Pfeiffer [4,5], Alexandre Bisson [2] ✉, Stefan Schulze [1,6] ✉ & Mechthild Pohlschroder [1] ✉

Archaea play indispensable roles in global biogeochemical cycles, yet many crucial cellular processes, including cell-shape determination, are poorly understood. *Haloferax volcanii*, a model haloarchaeon, forms rods and disks, depending on growth conditions. Here, we used a combination of iterative proteomics, genetics, and live-cell imaging to identify mutants that only form rods or disks. We compared the proteomes of the mutants with wild-type cells across growth phases, thereby distinguishing between protein abundance changes specific to cell shape and those related to growth phases. The results identified a diverse set of proteins, including predicted transporters, transducers, signaling components, and transcriptional regulators, as important for cell-shape determination. Through phenotypic characterization of deletion strains, we established that rod-determining factor A (RdfA) and disk-determining factor A (DdfA) are required for the formation of rods and disks, respectively. We also identified structural proteins, including an actin homolog that plays a role in disk-shape morphogenesis, which we named volactin. Using live-cell imaging, we determined volactin's cellular localization and showed its dynamic polymerization and depolymerization. Our results provide insights into archaeal cell-shape determination, with possible implications for understanding the evolution of cell morphology regulation across domains.

Archaea, although primarily known for their ability to thrive in extreme environments, are ubiquitous[1], play vital roles in a variety of ecological processes, and, as part of the human microbiome, likely have significant impacts on human health[2–7]. Yet, the mechanisms underpinning key aspects of archaeal cellular processes remain poorly understood, including those that regulate cell-shape determination.

Prokaryotic cells often assume specific shapes under distinct environmental conditions, indicating increased fitness benefits associated with the regulation of cell shape[8,9]. Specific shapes likely evolved as adaptations to improve nutrient acquisition, increase reproductive efficiency, facilitate resistance to antibiotics, colonize hosts effectively, and evade predation[8,10–12]. Cell-shape transitions may represent an evolutionary correlation between shape and lifestyle and can range from differentiation into specialized cells, such as in *Caulobacter crescentus*[13], to relatively simple and common morphological changes that occur when cells growing as rods during exponential phase shift to a spheroid shape upon entry into stationary phase, as exemplified by *Arthrobacter*, *Acinetobacter*, and *Rhodococcus equi*[14–16]. Overall, these cell-shape transitions occur alongside changing environments and appear to be under explicit genetic control.

[1]University of Pennsylvania, Department of Biology, Philadelphia, PA 19104, USA. [2]Brandeis University, Department of Biology, Waltham, MA 02453, USA. [3]University of Pennsylvania, Department of Physiology, Philadelphia, PA 19104, USA. [4]Biology II, Ulm University, 89069 Ulm, Germany. [5]Computational Biology Group, Max Planck Institute of Biochemistry, 82152 Martinsried, Germany. [6]Rochester Institute of Technology, Thomas H. Gosnell School of Life Sciences, Rochester, NY 14623, USA. ✉e-mail: bisson@brandeis.edu; sxssbi1@rit.edu; pohlschr@sas.upenn.edu

In bacteria, cell shape is often controlled through modulation of peptidoglycan in the cell wall[8,12], which can be achieved through various mechanisms. Helical cells require components that promote curvature, such as the polymer and filament-forming CrvA in *Vibrio cholerae*, which skews peptidoglycan synthesis rates[17], or the intermediate filament-like protein crescentin in *C. crescentus*[17,18]. Alternatively, the bacterial actin homolog MreB contributes to morphogenesis of many rod-shaped bacteria by controlling lateral peptidoglycan synthesis[19].

Archaeal species also exhibit diverse morphologies, ranging from rods[20] and cocci[21] to triangular or square-shaped cells[22–24]. While knowledge about archaeal cell-shape determinants is limited, the few components identified include the actin homolog crenactin in *Pyrobaculum calidifontis*, which is correlated with rod-shape in this species[25]. Moreover, a recently characterized Asgard archaeon *Candidatus* Lokiarchaeum ossiferum contains Lokiactin, a highly conserved actin homolog hypothesized to provide a scaffold for the maintenance of the archaeon's network of protrusions[26].

Some archaeal species change cell shape in response to specific environmental cues. For example, *Haloferax volcanii* is rod-shaped during early-log growth phase and when swimming[27–29]. Conversely, in mid- and late-log growth phases and at the center of motility halos, *Hfx. volcanii* forms flat, polygonal pleomorphic disks, presumably to maximize the cell surface for attachment or nutrient uptake[27–30]. However, while cell-shape transitions are consonant with significantly altered cell biology, few proteins that appreciably affect cell shape have been identified. *Hfx. volcanii* cells lacking archaeal tubulin homolog CetZ1 only make disks[29,30]. Conversely, cells depleted for the membrane protease LonB, which modulates CetZ1 degradation, predominantly form rods[31]. The deletion of the gene encoding the *Hfx. volcanii* peptidase archaeosortase A (ArtA) or either of the phosphatidylethanolamine biosynthesis enzymes, PssA and PssD, also results in rod-shaped predominance across growth phases[32,33].

Here, we report the isolation of *Hfx. volcanii* cell-shape mutants that, unlike previously identified morphology mutants, completely lack the ability to form disks. These rod-only mutants proved invaluable for proteomic comparisons to wild-type and disk-only cells at different growth stages. Based on differential protein abundances, we identify structural and regulatory components critical for rod and disk formation. Molecular genetics and cell-biological characterizations of selected candidates, including a distant actin homolog, support their involvement in cell-shape determination. Based on these results, we present a model for cell-shape determination in *Hfx. volcanii*.

## Results

### Hypermotility screens of a transposon library revealed *Hfx. volcanii* disk-defective mutants

A small number of *Hfx. volcanii* mutants have been reported to form predominantly, but not exclusively, rods under certain growth conditions[31–33]. Rods have been shown to be associated with motility, with the edges of motility halos containing rod-shaped cells and the rod-deficient mutant Δ*cetZ1* being non-motile[29], leading us to hypothesize that hypermotility may be associated with a disk-deficient phenotype. Therefore, we examined hypermotile mutants previously identified in a screen of a transposon insertion library[34,35] for cell shape. Out of five hypermotile strains identified through stab-inoculation screens[34], four formed predominantly rods, with some disks in later growth phases, but only one exclusively formed rods (Fig. 1): SAH1, which contains a transposon insertion upstream of *hvo_2176* (position 2,049,208)[34]. We repeated the screen and assessed the motility of an additional 1500 transposon insertion mutants, five of which were hypermotile. Three of the five formed predominantly rods, and only two of the five formed exclusively rods: JK3 and JK5 (Fig. 1). JK3 contained an insertion in *hvo_B0194* (plasmid pHV3, position 224,729),

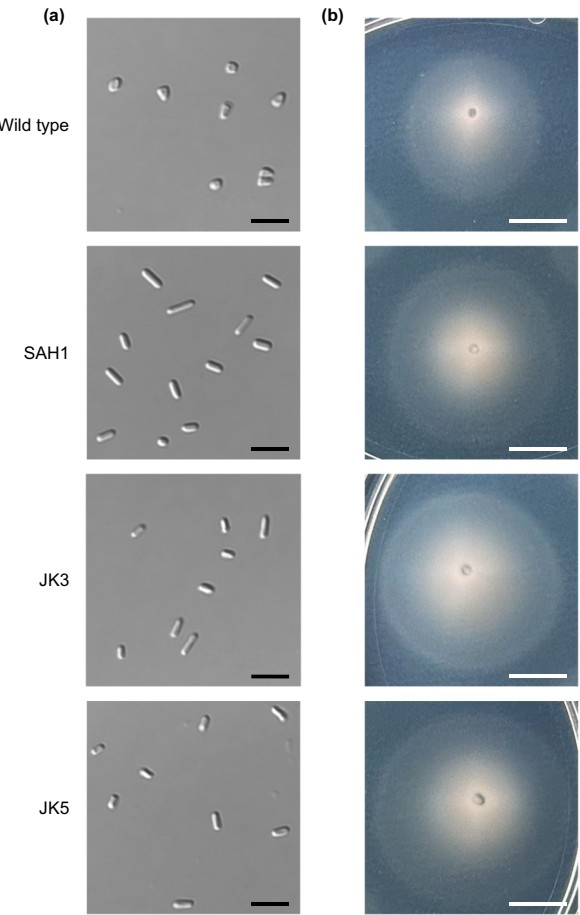

(a)                (b)

Wild type

SAH1

JK3

JK5

**Fig. 1 | Three hypermotile and rod-only mutants were isolated from hypermotility screens using a transposon library. a** Cell shape images using DIC microscopy at late log ($OD_{600}$ between 1.15 and 1.35) for wild type (H26), SAH1, JK3, and JK5 strains are shown, each representative of three biological replicates, with scale bars indicating 5 μm. **b** Images of motility halos for the same strains, each representative of three biological replicates, stab-inoculated in 0.35% agar plates and imaged after two days of incubation at 45 °C and one day at room temperature, are shown. Scale bars indicate 10 mm.

which encodes a hypothetical protein, and JK5 contained an insertion in *hvo_B0364* (plasmid pHV3, position 421,533), which encodes a molybdopterin oxidoreductase.

### Development of an interlinked quantitative proteomics experiment enabled focus on shape-specific effects

While screening hypermotile mutants for shape defects allowed us to identify genes potentially important for motility and shape, inherent limitations of genetic screens (Supplementary Note 1) prevent their use for more comprehensive analyses of shape-transition pathways. Therefore, we performed a system-wide analysis using quantitative proteomics to compare rod- and disk-shaped cells. Crucially, the availability of rod- and disk-only mutants allowed for an interlinked experimental setup (Fig. 2a), with three types of comparisons: (a) wild-type cells in early log versus late log, which represents a comparison between rods and disks, respectively, but includes confounding influences (e.g., metabolic differences) from growth phase-related changes, as suggested by previous analyses[36]; (b) wild-type rods versus disk-only mutants in early log; and (c) wild-type disks versus rod-only mutants in late log. The latter two were expected to yield shape-specific changes, and including comparisons at both growth stages allowed us to account for effects that may be specific to a distinct mutant or growth phase.

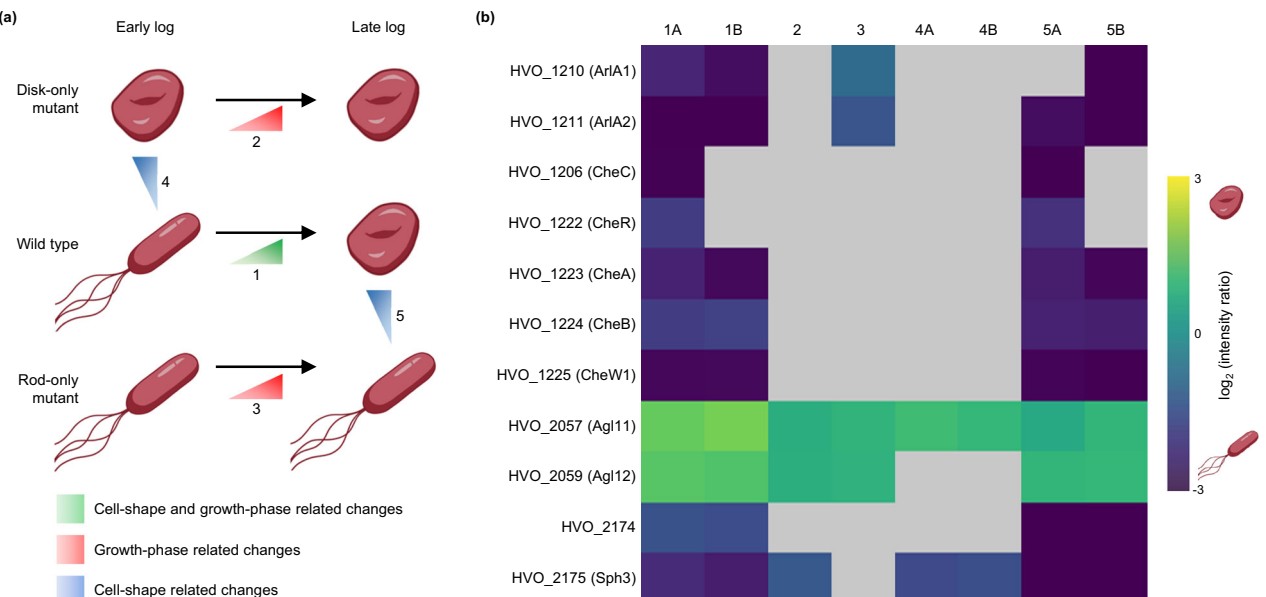

**Fig. 2 | A quantitative proteomics approach comparing wild-type and shape-defective mutants enabled the identification of proteins important for shape determination. a** Proteomics experiments were designed to compare disk- and rod-only mutants with wild type across early- and late-log growth phases. This interlinked experimental setup aimed to distinguish growth phase-related changes in protein abundances from cell-shape specific changes. **b** Results from label-free proteomics comparing wild type (H53 and H98), disk-only (ΔcetZ1), and rod-only (JK3) strains are shown as a heatmap, with colors indicating log₂-transformed intensity ratios (disk-forming conditions and strains as the numerator and rod-forming conditions and strains as the denominator). Positive and negative values indicate higher and lower abundances in disks, respectively. Ratios with a posterior error probability (PEP) < 0.05 are shown; otherwise, gray boxes are used. Each column in the heatmap corresponds to the comparison of two conditions or strains, with numbers representing the same comparisons as in **a**. Columns numbered with an A refer to the use of H53 as wild type, while those numbered with a B indicate the use of wild type H98. Columns 1A and 1B correspond to comparisons between early- and late-log growth phases for each wild-type strain, columns 2 and 3 correspond to comparisons between early- and late-log growth phases for disk- and rod-only mutants, respectively, and columns 4A, 4B, 5A, and 5B correspond to shape-specific comparisons. Raw data are provided in the Supplementary Data 1 file.

## Quantitative proteomics comparing JK3, ΔcetZ1, and wild-type cells across distinct growth phases revealed candidates for shape-specific components

We utilized the rod-only transposon mutant JK3 and disk-only deletion mutant ΔcetZ1[29] to be compared with wild-type strains H53 and H98 (see Supplementary Note 2 for the genealogy of strains) in early- and late-log growth phases. The main goal for this initial proteomics experiment was to determine whether the experimental design would facilitate identification of proteins involved in shape determination, subsequently allowing us to generate additional shape-defective gene deletion strains. These strains would be derived from a single parent strain rather than using transposon mutants that often have secondary mutations[34].

We were able to identify and perform label-free quantification of 1944 proteins; a differential abundance in at least one comparison could be found for 314 proteins (Supplementary Data 1). For most of these proteins ($n = 218$), significant changes in abundance were only seen for comparisons within a single strain across growth phases. However, 46 proteins showed differential abundances for comparisons between early- and late-log wild-type cells as well as at least one shape-specific comparison between wild-type and shape-mutant cells within a single growth phase (Supplementary Data 1). These proteins were further considered as potential candidates for proteins with a shape-related function.

The archaellin proteins ArlA1 and ArlA2 were highly abundant in rods as were chemotaxis proteins, including CheC, CheR, CheA, CheB, and CheW1 (Fig. 2b). Moreover, two components of the Agl15-dependent N-glycosylation pathway, Agl11 and Agl12, showed higher abundance in disk-forming conditions and mutants (Fig. 2b). Surprisingly, no differential abundances were observed for HVO_B0194; in line with these results, a deletion strain generated for hvo_B0194 did not

have a shape defect (Supplementary Fig. 1). These results suggest that other, secondary genome alterations in JK3 were responsible for the rod-only phenotype (Supplementary Table 1).

HVO_2174 and HVO_2175 were significantly more abundant in rods than in disks (Fig. 2b). Interestingly, they are genomic neighbors of hvo_2176, which was disrupted in the rod-only transposon mutant SAH1[34]. NCBI records indicate that HVO_2174 is a protein of unknown function, and HVO_2175 is annotated as Sph3, a structural maintenance of chromosomes (SMC)-like protein. When Halobacterium salinarum Sph1, a homolog of Sph3, is heterologously overexpressed in Hfx. volcanii, cells form rods[37]. These promising findings support the experimental design for the identification of shape-regulating candidates, and we next focused on the genomic region between hvo_2174 and hvo_2176.

## HVO_2174 and Sph3 are required for rod formation and wild-type-like motility

We individually deleted hvo_2174 and sph3 using parent strain H53 (ΔpyrE2, ΔtrpA)[38], referred to as wild type. Phenotypic analyses revealed that both Δhvo_2174 and Δsph3 formed only disks across all growth phases and were non-motile (Fig. 3). Upon complementation of Δhvo_2174 and Δsph3 cells with a plasmid overexpressing HVO_2174 and Sph3, respectively, the cells regained the ability to form rods (Supplementary Fig. 2). Neither strain had a growth defect (Supplementary Fig. 3). These shape data are consistent with the quantitative proteomics results and demonstrate the importance of HVO_2174 and Sph3 for rod formation. Thus, we propose to annotate HVO_2174 as rod-determining factor A (RdfA; gene rdfA).

Notably, analysis of HVO_2174 and Sph3 homologs revealed high conservation of synteny (Supplementary Table 2). RdfA defines a protein family (RdfA family) with a total of four paralogs in Hfx. volcanii

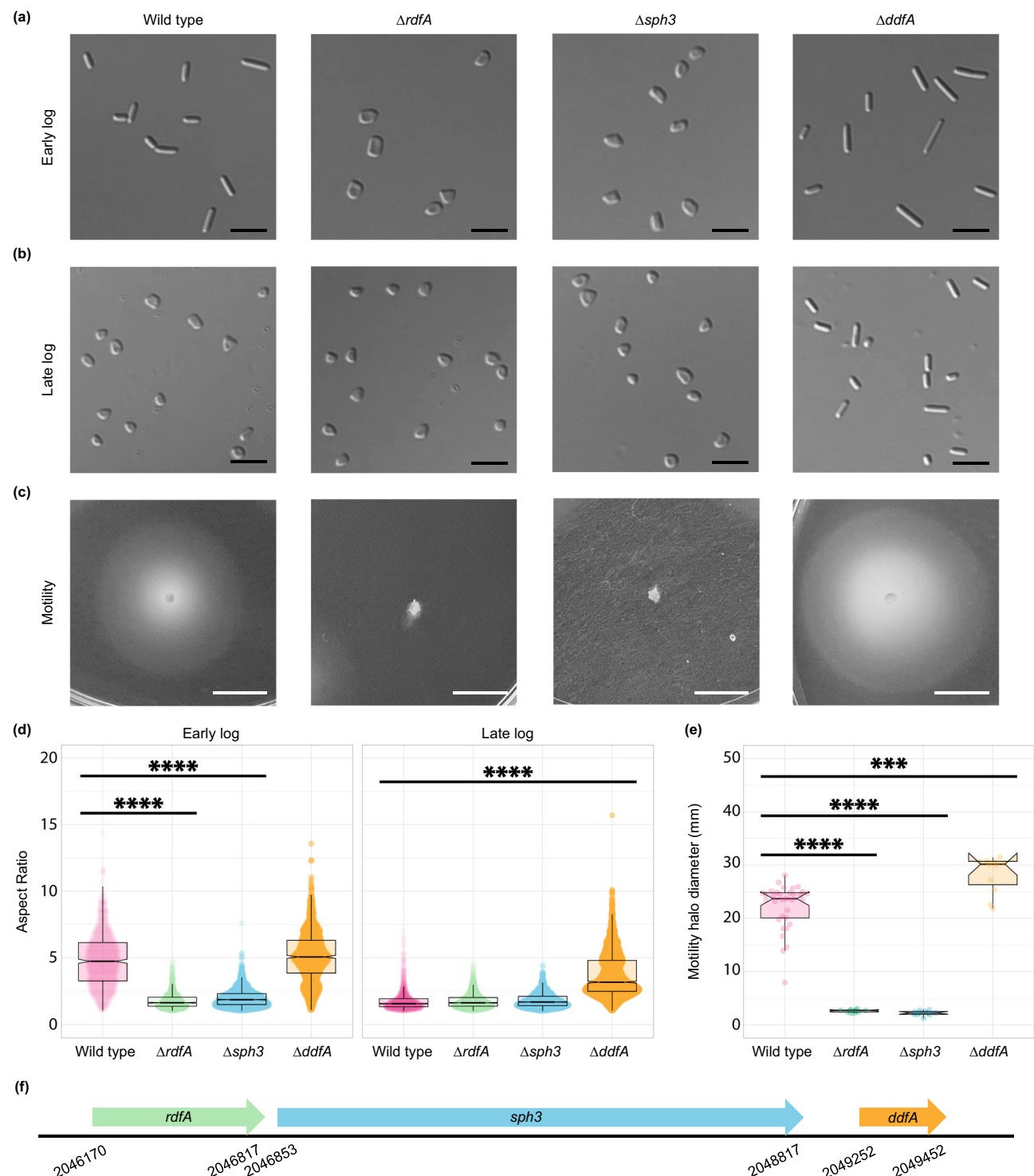

**Fig. 3 | RdfA, Sph3, and DdfA are required for wild type-like cell-shape determination and motility. a** Early-log and **b** late-log cell shape images for Δ*rdfA* (Δ*hvo_2174*), Δ*sph3* (Δ*hvo_2175*), and Δ*ddfA* (Δ*hvo_2176*) using DIC microscopy, each representative of three biological replicates, imaged at $OD_{600}$ 0.045 to 0.050 and 1.50 to 1.70, respectively. Rods in late log for Δ*ddfA* were a mix of elongated and short rods. Scale bars are 5 μm. **c** Motility halo images, representative of 33 biological replicates for wild type and 11 biological replicates for Δ*rdfA*, Δ*sph3*, and Δ*ddfA*. Strains were stab-inoculated in 0.35% agar plates and imaged after two days of incubation at 45 °C and one day at room temperature. Scale bars indicate 10 mm. **d** Quantification of aspect ratio for cell shape of each strain at early- and late-log growth phases. *n* for early log samples = 936, 835, 2698, and 1190 for wild type,

Δ*rdfA*, Δ*sph3*, and Δ*ddfA*, respectively; *n* for late log samples = 1339, 1358, 2431, and 2624 for wild type, Δ*rdfA*, Δ*sph3*, and Δ*ddfA*, respectively. Aspect ratio comparisons were assessed using an unpaired, nonparametric, two-tailed Kolmogorov-Smirnov test. ****$p < 0.0001$. Aspect ratios <2 are considered disks and/or short rods. **e** Quantification of motility halo diameter for each strain. df = 42. Halo diameters were assessed using an unpaired, two-tailed *t*-test. ****$p < 0.0001$, ***$p = 0.0001$. Boxplots show the mean (center line between boxes), interquartile range (boxes), 95% confidence interval of the mean (notch), as well as the lowest and highest 25th percentile of the distribution (whiskers). Source data are provided as a Source Data file. **f** Schematic representation of genomic region including *rdfA*, *sph3*, and *ddfA*.

(Supplementary Table 2). Analyzing 24 genomes from 19 haloarchaeal genera, 54 RdfA family proteins and 50 SMC-like proteins were identified. The majority of the RdfA family proteins (42 of 54, 78%) and the SMC-like proteins (42 of 50, 84%) are encoded directly adjacent to each other. There are only 12 RdfA family proteins and eight SMC-like proteins that lack a partner gene. In one of these cases, the two proteins are encoded in genomic vicinity with an intercalation of eight unrelated genes.

## HVO_2176 is required for disk formation

Given the rod-only phenotype of SAH1 as well as the requirement of *rdfA* and *sph3* for rod formation, we hypothesized that *hvo_2176* may also play a role in cell-shape determination. Indeed, after generating a deletion strain for *hvo_2176*, we found that Δ*hvo_2176* is hypermotile and forms only rods regardless of growth phase (Fig. 3). While Δ*hvo_2176* has a slight growth advantage in exponential phase, it reaches stationary phase at a similar time and $OD_{600}$ as wild type (Supplementary Fig. 3). We propose annotating HVO_2176 as disk-determining factor A (DdfA; gene *ddfA*). An attempt to complement Δ*ddfA* with a copy of *ddfA* carried on a plasmid was unsuccessful, leading us to characterize its gene annotation more carefully. Unlike most haloarchaeal DdfA homologs, the annotated *Hfx. volcanii* protein lacks approximately 126 nucleotides at the N-terminus. Notably, this annotation amendment places the SAH1 transposon insertion within, rather than upstream, of *ddfA*. Hence, we expressed DdfA with the traditional start codon ATG inserted upstream of its first attributable codon, GGA (*ddfA^ext*). Cells expressing the plasmid with *ddfA^ext* formed disks, confirming that DdfA is required for disk formation (Supplementary Fig. 4a–d). Subsequent transcriptional analysis revealed that the *ddfA* transcription start site may actually be located an additional 18 nucleotides upstream of the *ddfA^ext* start site, but there is no traditional start codon in the vicinity (Supplementary Fig. 4e). According to the Ribosome profiling data, the start codon could be GGG, located six nucleotides upstream of the *ddfA^ext* start (Supplementary Fig. 4e). However, the in vivo start codon of DdfA has yet to be identified.

While RdfA and Sph3 and their homologs are clustered across several genomes, DdfA is clustered with RdfA and Sph3 in *Hfx. volcanii* but is rarely clustered in other organisms.

## Comparative proteomics using rod-only Δ*ddfA* and disk-only Δ*rdfA* mutants across growth phases allowed for identification of additional shape-specific components

The validation of our quantitative proteomics results through reverse genetics provided a proof-of-concept that our experimental design is suitable to identify proteins involved in shape determination. Building on these results, we compared Δ*rdfA* and Δ*ddfA* with wild type in early- and late-log growth phases for a more comprehensive quantitative proteomics experiment. We further optimized the experiment using a more sensitive mass spectrometer and three rather than two biological replicates per condition to increase statistical power. This in-depth proteomic analysis resulted in the identification and label-free quantification of 2328 proteins, with 938 proteins showing differential abundances in at least one comparison (Supplementary Data 3).

In addition to knowledge-driven comparisons, we performed exploratory data processing using principal component analysis (PCA) and variance-sensitive clustering with VSClust[39] based on normalized abundances for each condition and strain. The PCA showed the strongest separation between growth phases, but differences between wild type and Δ*rdfA* samples from early log, as well as between wild type and Δ*ddfA* samples from late log, were observed as well, which is in line with phenotypes in the respective growth phases (Supplementary Fig. 5a). VSClust identified 14 clusters (Supplementary Fig. 5b–d; Supplementary Fig. 6; Supplementary Data 2) with corresponding protein abundance patterns for each condition and strain, allowing for

the selection of proteins that might function similarly or in the same pathways.

While most clusters indicated differences in abundance due to changes in growth phase (Supplementary Note 3), clusters 11 and 12 likely included proteins involved in shape determination (Fig. 4a, b). For cluster 11, protein abundances for wild type and rod-only Δ*ddfA* were lower than for disk-only Δ*rdfA* in early log (Fig. 4a) but similar across late log. Given that some Δ*ddfA* rods are shortened in late log, perhaps cells lacking *ddfA* initiate the transition to disks by forming short, rounded rods in late log, but DdfA is required to complete the transition to disks. We therefore hypothesize that cluster 11 comprises proteins that are involved in the early stages of disk transition. Besides metabolism-related proteins like the electron carrier halocyanin (HVO_0825), cluster 11 includes proteins that may have regulatory functions on the DNA and/or RNA level, such as Tfs2 (HVO_2632), a transcriptional elongation factor, and the zinc-finger protein HVO_1359.

Cluster 12 displays an opposing pattern to cluster 11, with abundances that were high for wild type and Δ*ddfA* in the early log but substantially lower for Δ*rdfA* in the early log and for all strains in late log; moreover, abundances for Δ*ddfA* in late log were higher than that of wild type and Δ*rdfA* (Fig. 4b). This abundance pattern for cluster 12 suggests that these proteins play important roles in rod formation. This cluster included RdfA as well as chemotaxis protein CheB (HVO_1224), transducers BasT (HVO_0554) and HemAT1 (HVO_1484), and transport system proteins HVO_1442, HVO_A0624, and HVO_3002. Regulatory elements present in cluster 12 include zinc-finger protein HVO_0758 and circadian clock protein CirD (HVO_2232).

The only cluster comprised of proteins with strikingly higher abundances in Δ*ddfA* than wild type and Δ*rdfA*, in both early log and late log, was cluster 14 (Fig. 4c). This cluster includes several proteins likely involved in genetic regulation, including transcriptional regulator TrmB1 (HVO_A0150), and the SMC-like protein Sph4 (HVO_B0173). Given the higher abundance of these proteins in the late log for the wild type, they might be important for initiating the transition to disks. The higher abundance in Δ*ddfA* in the early log and late log might be the result of cellular mechanisms compensating for increased rod formation in this mutant.

Along with the unbiased clustering approach, we performed knowledge-based filtering to identify proteins of interest (Supplementary Data 3). Filtering for proteins that showed differential abundances in comparisons for wild-type cells across growth phases, as well as between at least one shape mutant and wild type, resulted in 71 proteins/protein groups (Supplementary Data 3; Supplementary Fig. 7). Of these, 50 were not identified in this category in the initial proof-of-concept analysis, illustrating the benefits of using Δ*rdfA* and Δ*ddfA* mutants and the optimized mass spectrometry approach.

It is worth noting that the clustering was based on protein abundance patterns across all strains and conditions, and relatively small differences in these patterns, even for a single condition, can result in the exclusion of a protein from a cluster. In contrast, the knowledge-based filtering for proteins that showed differential abundances in shape-specific comparisons (Supplementary Fig. 7; Supplementary Data 1 and 3) can result in proteins with a variety of abundance patterns. Therefore, together, the clustering and knowledge-based filtering provide a more comprehensive picture of proteins that are likely involved in shape transitions.

For example, the transducers MpcT (HVO_0420) and Htr7 (HVO_1999), which showed higher abundances in rods relative to disks in comparisons between wild type and Δ*rdfA* (Supplementary Fig. 7), did not reach the significance threshold to be considered part of a cluster (Supplementary Data 2). However, their abundance pattern was most similar to that of proteins in cluster 12 (Fig. 4b), indicating that these proteins might be involved in signaling cascades that are not just

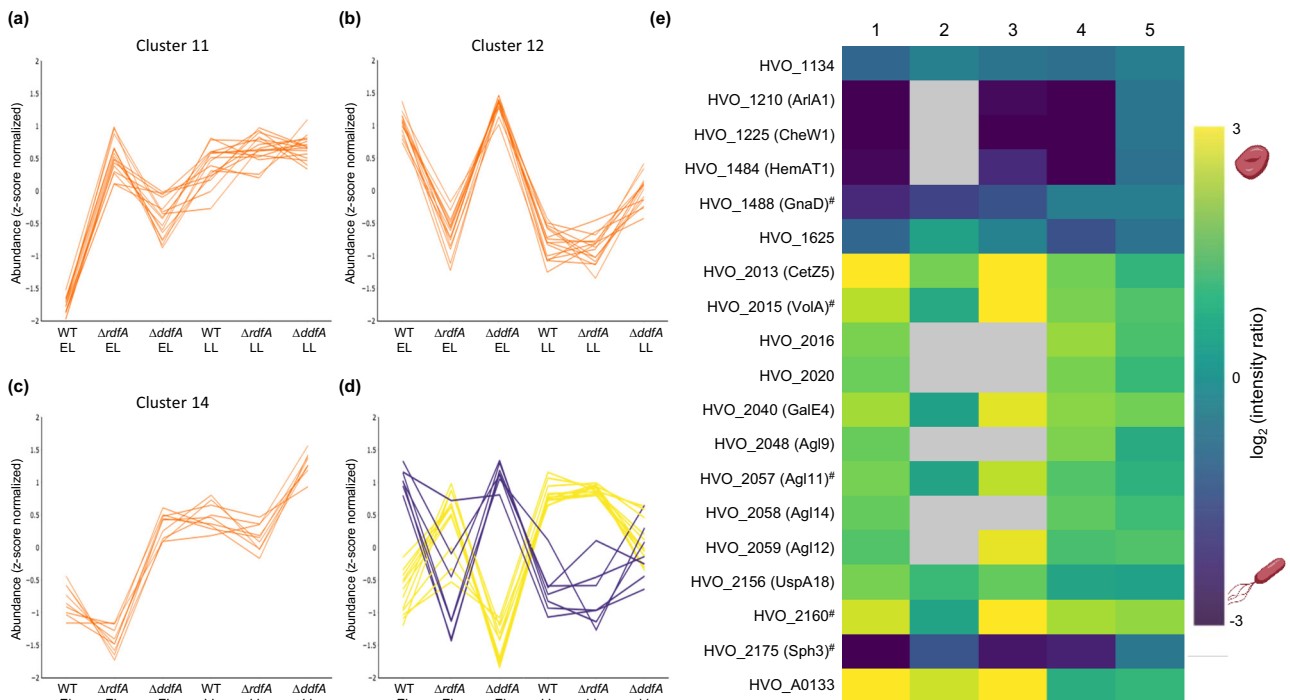

**Fig. 4 | Quantitative proteomics analyzing Δ*rdfA*, Δ*ddfA*, and wild type reveal additional proteins important for shape.** After performing variance-sensitive clustering based on protein abundances, clusters that exhibit patterns indicative for proteins that are likely involved in cell shape were selected: **a** cluster 11, **b** cluster 12, and **c** cluster 14 (for additional clusters, see Supplementary Fig. 6). Each line corresponds to an individual protein or protein group. Furthermore, protein abundance ratios were used to filter for proteins that exhibited significant differences (PEP < 0.05) in three shape-related comparisons: (1) wild type early- versus late-log growth phases, (4) Δ*rdfA* versus wild type at early log, and (5) wild type

versus Δ*ddfA* at late log. These proteins are represented by (**d**) a line plot of normalized abundances for proteins more abundant in disks (yellow) or rods (purple) and **e** a heatmap of log₂-transformed ratios. Column numbers correspond to the experimental design (see Fig. 2a), and ratios with PEP > 0.05 are shown as gray boxes. Proteins labeled with # also showed significant abundance differences for all three comparisons (columns 1, 4, and 5) for the proteomics experiment using JK3, Δ*cetZ1*, and wild type. WT: wild type, EL: early log, LL: late log. Raw data are provided in the Supplementary Data 2 and 3 files.

related to chemotaxis but also to shape transitions. Similarly, Sph3 and CetZ1 were only weakly associated with cluster 12 but showed significant abundance differences in shape-specific comparisons (Supplementary Fig. 7), and the phenotypes of their respective mutants Δ*sph3* and Δ*cetZ1*[29] confirmed their roles in shape determination.

In addition, various ABC transporters showed differential abundances in at least one shape-specific comparison (Supplementary Fig. 7). Two of them, HVO_1442 and HVO_3002, were part of cluster 12, and proteins encoded by their respective adjacent genes HVO_1443 and HVO_3001 showed similar abundance patterns (Supplementary Data 2). However, the ABC transporter HVO_1705, predicted to transport iron-III[40], was part of cluster 4 (Supplementary Data 2), indicating that its function might be related to changes in the growth phase rather than shape. Other ABC transporters (HVO_2031, HVO_2163) were not clearly associated with a specific cluster (Supplementary Data 2). The substrates for many of these transporters remain to be elucidated.

Our analyses also indicated some proteins of unknown function to be involved in rod formation, such as HVO_1134, which was a member of cluster 12 and also selected in the knowledge-based comparisons of wild type and shape mutants. While its exact role remains to be elucidated, it is annotated as a transcriptional regulator by ProtNLM.

Another notable observation from combining the clustering and knowledge-based approaches was the identification of three genomic regions containing genes encoding a few to several proteins with suspected importance in shape determination. These regions include *hvo_2160* to *hvo_2175*, *hvo_2046* to *hvo_2061*, which encodes many Agl15 *N*-glycosylation pathway proteins, and *hvo_2013* to *hvo_2020* (Supplementary Note 4).

To determine a list of proteins with the strongest evidence for involvement in shape determination, we focused on those that showed differential abundance across all three shape-relevant comparisons between wild type, Δ*rdfA*, and Δ*ddfA*. Through this filtering, 19 proteins of interest (Fig. 4d, e) were selected, five of which also showed significant abundance differences in proteomic comparisons with JK3 and Δ*cetZ1* (Fig. 4e). The abundance patterns for each of these 19 proteins resembled either cluster 11 or 12 (Fig. 4d), in line with their likely importance in shape determination. It should be noted that the initial *rdfA* deletion strain (HS37) used for the proteomics had retained part of the pTA131 plasmid. However, the cell shape phenotypes of this Δ*rdfA* strain (Fig. 3) and the subsequent deletion strain of *rdfA* (HS50) devoid of pTA131, used for the complementation experiments (Supp. Fig. 2), are the same.

## HVO_2015, an actin homolog, is important for disk-shape formation

HVO_2015 was one of five proteins that showed the strongest evidence to be involved in shape, based on proteomic comparisons, and is likely to play a structural role in disk formation. Structural and sequence analysis as well as BLAST searches revealed that HVO_2015 is similar to actin proteins (Supplementary Fig. 8a, b; Supplementary Note 5), providing strong evidence for a likely cytoskeletal function. We annotated HVO_2015 as volactin (VolA; gene *volA*) in reference to the organism in which it was identified.

To elucidate the role of volactin, we attempted to generate a Δ*volA* strain. However, of 100 colonies tested for a full deletion, we could only isolate a partial deletion of this polyploid haloarchaeon, which retained a few gene copies (denoted as Δ*volA**) (Supplementary

Fig. 9a, b), suggesting that volactin is essential. Mid-log Δ*volA*\* cultures contained significantly more rods relative to wild-type cultures (Fig. 5a), indicating that volactin is important for rod-to-disk shape transitions. However, as Δ*volA*\* cultures transitioned to late log, they formed disks like wild type. Furthermore, Δ*volA*\* exhibits motility similar to the parent strain (Supplementary Fig. 8c) and does not have a growth defect (Supplementary Fig. 3). The shape defect was corrected upon complementation of Δ*volA*\* with a plasmid containing VolA (Supplementary Fig. 8e). These results support a structural role for volactin in disks and indicate that, while shape and motility may have common regulators in some contexts, they nevertheless represent at least partially independent processes, with distinct proteins involved in each process.

### Volactin assembles into dynamic filaments in vivo and is highly abundant in disks

We next tested whether volactin can polymerize in vivo as expected for a cytoskeletal protein. We generated a knock-in strain replacing *volA* with a *volA-msfGFP* construct under control of its native promoter and determined that the effect of this replacement on shape was minimal (Supplementary Fig. 8). Using epifluorescence microscopy, we observed the assembly of long filaments across the cell and round patches at the membrane (Fig. 5b). To determine where volactin filaments are located relative to the cytokinetic ring, we transformed the *volA-msfGFP* construct into a strain expressing an FtsZ1-mApple fusion. Colocalization studies confirmed that volactin filaments assemble and disassemble independent of the division site (Fig. 5b; Supplementary Movie 1).

The assembly of polymers and patches of different sizes and locations in the cell are a hallmark of spatiotemporal organization. To inspect their turnover timeline, we tracked cells over one doubling time (Fig. 5c). Remarkably, we observed a diversity of dynamic behaviors, including polymer elongation, fast depolymerization, and appearance and disappearance of patches. We also frequently observed long filaments connecting sibling cells at the onset of cell division. In these cases, filaments rapidly depolymerize right before completion of cytokinesis, suggesting that polymerization and depolymerization events may be carefully regulated, perhaps by as-yet unknown volactin-binding factors.

To better understand the assembly of patch-like structures, we examined volactin polymers in 3-D projections to determine where filaments were placed relative to the membrane and cytoplasm. Strikingly, super-resolution projections of cells expressing volactin-msfGFP showed that filaments never were adjacent to the membrane but rather bound the membrane by their tips, bridging the cytoplasm (Fig. 5d; Supplementary Movie 2). Moreover, the apparent patches were actually a top view of this volactin-tip anchoring (Fig. 5d; Supplementary Movie 2). In conclusion, volactin exhibits a different dynamic behavior compared to bacterial MreB[41,42], closely resembling dynamically unstable actin polymers as exemplified by ParM[43,44].

Given the role of volactin in morphological differentiation, and based on our proteomic data suggesting higher volactin levels in disks, we expected volactin filaments to be more active in disks compared to rods. Analyzing images of cells expressing volactin-msfGFP at early- (primarily rods) and mid-log (mixed rods and disks) growth phases, we observed volactin filaments in both populations (Fig. 5e) but a higher fluorescent signal from volactin-msfGFP polymers in mid-log compared to early-log cells which we attributed to a higher number of disks in mid log (Supplementary Fig. 8f; Supplementary Note 5). These results, combined with the Δ*volA*\* morphology phenotype, reveal volactin as part of a cytoskeletal system involved in disk-shape formation.

## Discussion

In this work, we combined transposon insertion screens and quantitative proteomics with reverse genetics and advanced microscopy to identify and characterize proteins crucial for cell-shape determination in *Hfx. volcanii*. These include structural and regulatory proteins linked to cell-shape transitions, ranging from filaments and *N*-glycosylation pathway enzymes to genetic regulators and transducers (Fig. 6). Underscoring the success of our multifaceted approach, the genetic screens and subsequent reverse genetics revealed *ddfA* as a key player in disk formation, while the corresponding protein was not identified by proteomics, likely due to its small size. Conversely, Sph3 and volactin were identified using quantitative proteomics as proteins involved in rod- and disk-formation, respectively, with knockouts and microscopic analysis providing confirmation. Transposon insertion screens would not have identified volactin since the gene encoding it appears to be essential. The use of fluorescence microscopy allowed us to expand upon the results from the system-wide proteomics, as we gained insights into the formation of volactin filaments.

Of the proteins implicated for shape determination, many transducers appeared to be associated with rod formation. Transducers sense and respond to a variety of environmental stimuli and are believed to coordinate signal transduction with the chemotaxis machinery[45]. In *Hbt. salinarum* and *Hfx. volcanii*, transducers relay distinct signals into the core of the signal transduction cascade, CheA and CheY[45]. Since *Hfx. volcanii* motility correlates with rods, the higher abundance of archaellins, chemotaxis proteins, and transducers in rods matches our expectations. In addition, it is tempting to speculate that elements of the signal relay cascade of the chemotaxis system are also involved in initiating shape transitions. Associated with these cascades, the distinct ABC transporters with differential abundance in rods and disks may be involved in transporting signals across the membrane.

We also determined that the genomic region spanning *hvo_2160* to *hvo_2176*, including *rdfA*, *sph3*, and *ddfA*, which were characterized using reverse genetics, is important for shape regulation. DdfA, a hypothetical protein, contains a halobacterial output domain 1 (HalOD1), which has been associated with signal transduction pathways[46], suggesting a potential role in signal cascades that regulate cell-shape determination. While the specific interactions of DdfA are unresolved, the possibility that DdfA interacts with RdfA and Sph3, of which homologs are frequently co-localized across genomes (Supplementary Table 2), is intriguing.

A BLAST search using RdfA revealed mostly additional hypothetical haloarchaeal proteins but also identified a *Halorussus* M15 family metallopeptidase (WP_163523357.1). In *Helicobacter pylori*, a metallopeptidase is required to maintain its helical shape[47]. Sph3 is an SMC-like protein, which in prokaryotes typically form homodimers associated with chromosome segregation and DNA aggregation[48,49]. However, the cytoskeletal protein Arcadin-4, an SMC-like protein, associates with crenactin and, together, these play a role in cell-shape determination[25]. Moreover, a recent study revealed the formation of filaments consisting of two proteins with SMC domains in an *Anabaena* species, allowing it to maintain its multicellular structure[50]. These *Anabaena* proteins with SMC domains are coiled-coil-rich proteins (CCRPs)[50], which, in general, perform similar functions to eukaryotic intermediate filaments[19]. Intermediate filament-like proteins and CCRPs play important roles in morphology, motility, and growth[19], suggesting a potential cytoskeletal-like role for SMC-like proteins. Thus, Sph3 may play a similar role in promoting rod formation in *Hfx. volcanii*. Another SMC-like protein, Sph4, was also identified to be likely involved in cell-shape determination, as a member of cluster 14, pointing to a potential general cell shape effect of SMC-like proteins.

We also identified volactin, a conserved but previously unannotated actin homolog important for the transition to disks. Actin proteins are found in all three domains of life and are involved in a variety of cellular processes. Yet, actins appear to be poorly distributed among archaea[21,51,52]. Curiously, despite resembling bacterial MreB (Supplementary Fig. 8a), our data show that volactin filaments are not

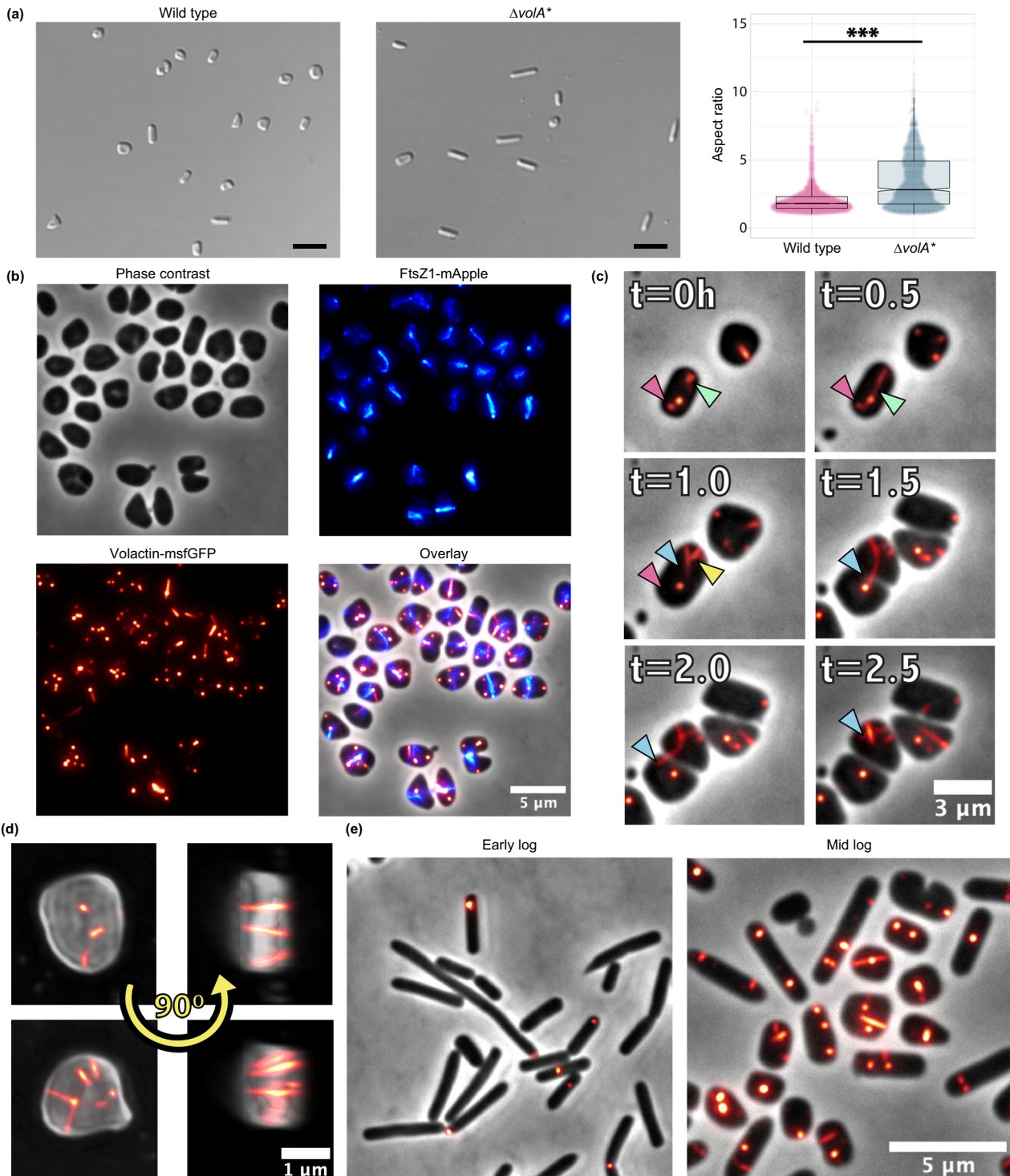

**Fig. 5 | Volactin is required for timely transition to disks and forms dynamic filaments in the cell. a** Cell shape images using DIC microscopy for wild type and Δ*volA\** at mid log (OD$_{600}$ 0.3), representative of three biological replicates, and quantification of aspect ratio. $n$ = 2804 and 1668 for wild type and Δ*volA\**, respectively. Aspect ratio comparisons were assessed using an unpaired, non-parametric, two-tailed Kolmogorov-Smirnov test. \*\*\**p* < 0.001. Aspect ratios <2 are considered disks and/or short rods. Scale bars are 5 μm. **b** Fluorescent microscopy of tagged VolA and FtsZ1 as well as overlay of the expression of each. Data is representative of three biological replicates. **c** Dynamic behaviors tracked over one doubling time, including polymer elongation (green arrowhead), fast

depolymerization (yellow arrowhead), and the appearance and disappearance of patches (magenta arrowhead). Long filaments connecting sibling cells at the onset of cell division are shown by a blue arrowhead. Data is representative of three biological replicates. **d** 3-D projections of cells expressing GFP-tagged volactin polymers. Data is representative of three biological replicates. **e** Localization of GFP-tagged volactin in early log (rods) and mid log (rods and disks). Data is representative of three biological replicates. Boxplots show the mean (center line between boxes), interquartile range (boxes), 95% confidence interval of the mean (notch), as well as the lowest and highest 25th percentile of the distribution (whiskers). Source data are provided as a Source Data file.

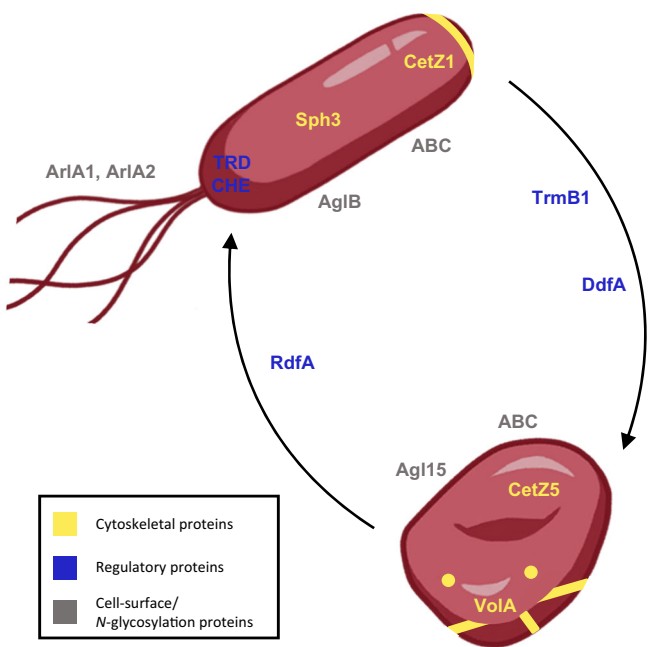

**Fig. 6 | Proposed model for structural and regulatory components of *Hfx. volcanii* cell-shape determination.** Proteins in yellow are predicted or likely cytoskeletal elements, those in blue are predicted regulatory elements, and those in gray are cell-surface proteins and enzymes required for *N*-glycosylation. TRD is transducers; CHE is chemotaxis machinery proteins; ABC is ABC transport systems; AglB represents the AglB-dependent *N*-glycosylation pathway; Agl15 represents the Agl15-dependent *N*-glycosylation pathway.

involved in rod-shape determination; on the contrary, low volactin protein abundance delays rod-to-disk shape transitions (Fig. 5a). The observation that rods at mid log show significantly higher volactin polymer signal compared to rods at early log (Fig. 5e; Supplementary Fig. 8f) suggests that volactin is an early molecular marker for the onset of transition from rods to disks.

Volactin is not the only factor known to be involved in the timely disk transition. Depletion of enzymes such as archaeaosortase A (ArtA), phosphatidylserine synthase (PssA), decaboxylase (PssD), and the protease LonB all result in a larger proportion of rod-shaped cells than wild type in mid log[33,53]. Although ArtA, PssA, and PssD were shown to localize with the cytokinetic ring[33], future studies should explore whether volactin filaments are controlling proteolysis and development in space and time. Likewise, it will be important to understand whether substrates are transported along the volactin filaments, which span the cytoplasm from one membrane region to another, and if so, what these substrates are.

Concomitantly with our work, salactin, another actin filament system, was characterized in the haloarchaeon *Hbt. salinarum*[54]. Salactin polymers are dynamically unstable, similar to our observations for volactin (Fig. 5c; Supplementary Movie 1), but salactin filaments are anchored at the cell poles and polymerize towards the cytoplasm. Intriguingly, unlike volactin, salactin plays no role in cell morphogenesis but rather aids in DNA partitioning when chromosomal copy number is low. The identification of these two functionally distinct classes of halophilic actins suggests that additional archaeal cytoskeletal proteins may be dedicated to very specific cellular processes.

We have identified both regulatory and structural elements that play roles in cell-shape determination, as outlined in Fig. 6. Deletion strains and live-cell microscopy not only confirmed the importance of selected components for shape determination but also provided further insights into their functions. Similar approaches could be used to determine the roles of the various proteins that were additionally

identified. Overall, the work presented here shows that complementary approaches can enable a deeper understanding of cellular processes, such as archaeal cell-shape determination.

## Methods

### Growth conditions
The analyzed mutants are based on distinct parent strains. The full list of parent and mutant strains can be found in Supplementary Table 1. All strains are derived from the same ancestral strain (H26) and differ by only a few genes. Details are provided in Supplementary Note 2. The *Hfx. volcanii* H53 wild-type strain and the mutant strains Δ*rdfA*, Δ*ddfA*, Δ*hvo_BO194*, Δ*sph3*, and Δ*volA\** were grown at 45 °C in liquid (orbital shaker at 250 rpm, 1-in orbital diameter) or on solid agar (containing 1.5% [wt/vol] agar) semi-defined Casamino Acids (Hv-Cab) medium[27], supplemented with tryptophan (+Trp) (Fisher Scientific) and uracil (+Ura) (Sigma) at a final concentration of 50 μg mL⁻¹. Ura was left out for strains carrying pTA963 and pTA963-based complementation plasmids. The *Hfx. volcanii* H26 wild-type strain and transposon mutant strains JK3, JK5, and SAH1 were grown with Ura supplementation, and Trp supplementation was not required. The *Hfx. volcanii* Δ*cetZ1* deletion strain[29] and its parent strain H98 were grown accordingly except that Trp supplementation was not required, but each had supplementation with hypoxanthine (+Hpx) (Sigma) and thymidine (+Thy) (Acros Organics), each at a final concentration of 40 μg mL⁻¹[55]. *Escherichia coli* strains were grown at 37 °C in liquid (orbital shaker at 250 rpm, 1-in orbital diameter) or on solid agar (containing 1.5% [wt/vol] agar) NZCYM (RPI) medium, supplemented, if required, with ampicillin at a final concentration of 100 μg mL⁻¹.

### Motility assay with transposon library to identify hypermotile mutants
Transposon mutants JK3, JK5, and SAH1 were identified through individual motility stabs of the transposon library and subsequent screening for hypermotility. These screens were performed as previously described[34]. Briefly, single colonies of the transposon library were stabbed into 0.35% motility agar plates, incubated for two to four days at 45 °C, and then assessed for motility, with the colonies that had halos at least 1.3 times the radius of wild type selected. Selected cells from the edge of these motility halos were then streaked on an agar plate and re-checked for hypermotility. Whole genome sequencing was performed on isolated transposon mutants via Illumina sequencing performed by SeqCenter (Pittsburgh, PA, USA).

### Sample collection for quantitative proteomics of wild type and shape-defective mutants across different growth phases
For the comparative proteomic analysis of wild-type strains H53 and H98 and shape-defective strains Δ*cetZ1*[29] and JK3, samples were collected at both early- and late-log growth phases in biological duplicates. Individual colonies for each strain were inoculated in 5 mL of liquid medium and incubated at 45 °C. Once early-log cultures reached an OD₆₀₀ between 0.045 and 0.055, an aliquot of each culture was subjected to a light microscopy morphology assay to verify that the shape was in accordance with expectations at the corresponding growth phase. At the same time, an aliquot of 4 mL was centrifuged at 6000 *g* for 10 min. The supernatant and pellet were separated, and pellets were frozen until further processing. To minimize protein degradation, 4-(2-aminoethyl)benzenesulfonyl fluoride HCl (AEBSF) (ThermoScientific) was added to the supernatant to a final concentration of 1 mM and gently vortexed. The supernatant was then concentrated to 100–200 μL using centrifugal filter units with a 3-kDa molecular mass cutoff (Amicon Ultra Centrifugal Filters, 4 mL, Merck Millipore) by centrifugation at 5000 *g* at 4 °C. Salts were removed by washing with 2 mL of double-distilled water (ddH₂O) and centrifugation at 5000 *g* at 4 °C. The concentrated solution (100–200 μL) was transferred to an Eppendorf tube, and the filter membrane was rinsed

with 100 μL ddH$_2$O to minimize protein loss, transferring the solution after rinsing to the same Eppendorf tube. The sample was then frozen at −80 °C until further processing.

For late-log cultures, the same steps were performed with the following changes: samples were collected (and cell morphology was verified) at an OD$_{600}$ between 1.0 and 1.2. Due to the higher cell density and protein concentration, the aliquot volume was reduced to 2 mL. After separation of pellets and supernatant, the supernatant from late-log cultures was concentrated to <100 μL and washed (with 400 μL of ddH$_2$O) in 3-kDa molecular mass cutoff (Amicon Ultra Centrifugal Filters, 4 mL, Merck Millipore) using centrifugation at 5000 g.

For the comparative proteomic analysis of H53, Δ*rdfA*, and Δ*ddfA*, the sample collection at both early- and late-log growth phases was optimized using biological triplicates along with the following modifications to the protocol. Individual colonies for each strain were inoculated in 1 mL liquid medium and vortexed to create an initial inoculum. The OD$_{600}$ of the initial inoculum was measured using a BioTek PowerWaveX2 microplate spectrophotometer with BioTek Gen5 v.1.11.5 (Agilent). Aliquots from the initial inoculum of each strain were added to 5 mL of liquid medium such that each culture started with a similar cell density, and cultures would reach the appropriate optical densities at similar times. Cultures were then incubated at 45 °C. Early- and late-log cultures within the same strain and replicate originated from the same initial inoculum. Samples for early-log cultures were taken once they reached an OD$_{600}$ between 0.045 and 0.050, while samples for late-log cultures were taken at an OD$_{600}$ between 1.7 and 1.8. The aliquot volume used for late-log cultures was 0.5 mL, and after separation of pellets and supernatant, the supernatant from late-log cultures was concentrated to <100 μL and washed (with 400 μL of ddH$_2$O) in 0.5 mL centrifugal filter units (3-kDa molecular weight cutoff, Millipore), using centrifugation at 14,000 g. For both growth phases, an aliquot of each culture was subjected to a light microscopy morphology assay to verify that the shape was in accordance with expectations at the corresponding growth phase and strain. All subsequent sample processing was performed as described for the other strains above.

### Protein extraction and preparation of samples for mass spectrometric analysis

Samples from all strains and conditions were treated equally for protein extraction and mass spectrometric sample preparation. Cells were lysed by thawing the cell pellet on ice and adding 400 μL of ddH$_2$O with 1 mM AEBSF. The lysate was transferred into a 0.5 mL filter unit (3-kDa molecular weight cutoff, Millipore) and concentrated to <100 μL by centrifugation at 14,000 g. The corresponding concentrated culture supernatant (see above) was added to the concentrated cell lysate. The solution was concentrated and washed two times with 400 μL ddH$_2$O, with centrifugation at 14,000 g in between the individual steps. To solubilize proteins, 400 μL of 2% SDS in 100 mM Tris/HCl, pH 7.5 was added, followed by incubation at 55 °C for 15 min. Afterwards, the solution was washed two more times with 400 μL ddH$_2$O (centrifugation at 14,000 g in between the individual steps). The resulting whole proteome extract was transferred to an Eppendorf tube, and the filter membrane was rinsed two times with 30 μL ddH$_2$O to minimize protein loss, transferring the solution after rinsing to the same Eppendorf tube. Protein concentrations were determined using a BCA assay following manufacturer's instructions (ThermoScientific).

For each sample, 20 μg of proteins were processed, including reduction with DTT and alkylation using iodoacetamide, and digested with trypsin using S-trap mini spin columns (ProtiFi) following manufacturer's instructions, with tryptic digestion overnight at 37 °C, as established in previous analyses[56]. Resulting peptides were dried in a vacuum centrifuge, and afterwards desalted using homemade C18 stage tips (3 M Empore Discs, CDS Analytical) as previously

described[57]. Briefly, stage tips were first conditioned with 100 μL CAN, then equilibrated with 100 μL 0.1% trifluoroacetic acid (TFA) twice, before loading the resuspended (in 0.1% TFA) peptides. Three washes with 100 μL 0.1% TFA were performed, after which peptides were eluted twice with 80 μL 0.1% formic acid in 60% CAN. Peptides were dried again in a vacuum centrifuge and stored at −20 °C until mass spectrometric analysis.

### Mass spectrometry of protein samples

Peptides were reconstituted in 0.1% formic acid (solvent A), and peptide concentrations for each sample were estimated based on absorbance measurements at 280 nm (NanoDrop™, Thermo Fisher Scientific). For each sample, 330 ng of peptides were chromatographically separated on a C18 column directly linked to an orbitrap mass spectrometer. The specific instruments, gradient conditions, and mass spectrometry parameters for the different samples are listed in Supplementary Table 3.

### Bioinformatic analyses of mass spectrometry for the identification and quantification of peptides and proteins

For the identification of peptides and proteins from the mass spectrometric data, the bioinformatic workflow used by the Archaeal Proteome Project[58] was employed. All analyses were performed within the Python framework Ursgal (version 0.6.9)[59] unless otherwise stated. Briefly, for the conversion of mass spectrometry raw files into mzML and MGF format, the ThermoRawFileParser (version 1.1.2)[60] and pymzML (version 2.5.0)[61] were used, respectively. Protein database searches with MSFragger (version 3.0)[62], X!Tandem (version vengeance)[63], and MS-GF+ (version 2019.07.03)[64] were performed against the theoretical proteome of *Hfx. volcanii* (4,074 proteins, version 190606, June 6, 2019, https://doi.org/10.5281/zenodo.3565631), supplemented with common contaminants, and decoys using shuffled peptide sequences. Mass spectrometry runs were calibrated and search parameters were optimized using MSFragger, resulting in a precursor mass tolerance of +/−10 ppm, and a fragment mass tolerance of 15 ppm. Oxidation of M and N-terminal acetylation were included as potential modifications, while carbamidomethylation of C was required as a fixed modification. Otherwise, default parameters were used. Peptide spectrum matches (PSMs) were statistically post-processed with Percolator (version 3.4.0)[65], and results from different search engines were combined using the combined PEP approach as described before[66]. After the removal of PSMs with a combined PEP > 1%, results were sanitized, i.e., for spectra with multiple differing PSMs, and only the PSM with the best combined PEP was accepted.

FlashLFQ (version 1.1.1)[67] was used for label-free peptide and protein quantification, with the match-between-run option activated (retention time window: 0.5 min), and performing a Baysian fold change analysis with a fold-change cutoff of 0.1. Based on the performed comparisons between wild-type and shape-mutant strains across different growth conditions, the strain or condition that served as the denominator for the comparison was specified as the control condition for the Bayesian fold change analysis. Log$_2$-transformed fold changes were plotted as heatmaps using Plotly (https://plotly.com), applying a PEP threshold of 5%.

In addition, variance-sensitive fuzzy clustering was performed using VSClust (version 1.2)[39]. As input for VSClust, protein abundances reported by FlashLFQ were z-transformed, and all proteins with six or more missing values across the 18 samples (six strains and conditions with three biological replicates each) were removed. Cluster numbers were estimated using the minimal centroid distance and Xie-Beni index[68] implemented in VSClust (Supplementary Fig. 5b−d), and 14 clusters were chosen for the variance-sensitive clustering. Furthermore, a principal component analysis was part of the VSClust workflow.

## Plasmid preparation and *Hfx. volcanii* transformation

The pTA131 plasmid was used to generate chromosomal deletions[38]. DNA Phusion Taq polymerase, restriction enzymes, and DNA ligase were purchased from New England BioLabs. Plasmids were initially transformed into *E. coli* DH5α cells; plasmids were then transformed into *E. coli* Dam⁻ strain DL739 prior to *Hfx. volcanii* transformation. Plasmid preparations for both *E. coli* strains were performed using the PureLink™ Quick Plasmid Miniprep Kit (Invitrogen). *Hfx. volcanii* transformations were performed using the polyethylene glycol (PEG) method[55]. All oligonucleotides used to construct the recombinant plasmids are listed in Supplementary Table 1.

## Generation of chromosomal deletions in *Hfx. volcanii* H53

Chromosomal deletions were generated by the homologous recombination method (pop-in/pop-out), as previously described[38] and outlined below. Plasmid constructs for use in the pop-in/pop-out deletion process were generated by using overlap PCR, as previously described[69]: about 750 nucleotides flanking the corresponding gene were amplified with PCR (for primers see Supplementary Table 1) and cloned into the haloarchaeal suicide vector pTA131[38]. For *rdfA*, the upstream flanking region was amplified with oligonucleotides 2174_KO_up_fwd and 2174_KO_up_rvr, and the downstream flanking region was amplified with 2174_KO_down_fwd and 2174_KO_down_rvr. The *rdfA* upstream and downstream flanking DNA fragments were fused by PCR using oligonucleotides 2174_KO_up_fwd and 2174_KO_down_rvr followed by cloning into the pTA131 vector digested with XbaI and XhoI. The resulting construct was verified by sequencing using the same oligonucleotides. The final plasmid construct contained upstream and downstream *rdfA* flanking regions and was transformed into the parental strain H53. Pop-out transformants were selected on agar plates containing 5-fluoroorotic acid (FOA) (Toronto Research Chemicals Inc.) at a final concentration of 50 μg mL⁻¹. The equivalent protocol was used for generation of the deletion or partial deletion strains Δ*ddfA*, Δ*hvo_B0194*, Δ*sph3*, and Δ*volA\**. Primers for the generation of these deletions are listed in Supplementary Table 1. Successful gene deletion (or partial deletion for Δ*volA\**) was confirmed by PCR using primers flanking the gene itself and/or primers that bind further upstream and downstream of the gene. The deletion strains Δ*rdfA*, Δ*sph3*, and Δ*ddfA* were further analyzed by Illumina whole genome sequencing performed by SeqCenter (Pittsburgh, PA, USA) to confirm the complete deletion of the gene as well as search for any secondary genome alterations. The Illumina files for the whole genome sequencing can be found at https://doi.org/10.5281/zenodo.8404691.

## Construction of expression plasmids in *Hfx. volcanii* H53

To construct tryptophan-inducible genes encoding *Hfx. volcanii* RdfA, Sph3, DdfA, and volactin with a C-terminal His tag, its coding region was amplified by PCR. For RdfA, Sph3, and volactin, the coding region was amplified using the oligonucleotides 2174_OE_fwd/2174_OE_rvr, 2175_OE_fwd/2175_OE_rvr, and 2015_OE_fwd/2015_OE_rvr, respectively (Supplementary Table 1). For DdfA, the coding region was amplified using the oligonucleotides 2176_OE_fwd_newannot and 2176_OE_rvr (Supplementary Table 1). Amendment of the annotation of *ddfA* (*ddfA^ext*) required designing a new forward primer for the expression plasmid (2176_OE_fwd_newannot) that began 126 nucleotides upstream of the annotated start site; additionally, a traditional ATG start codon was added in front of the first codon. The original forward primer used is listed as 2176_OE_fwd in Supplementary Table 1. The PCR products for RdfA, Sph3, DdfA, and volactin were then cloned into the expression vector pTA963, which had been digested with NdeI/EcoRI restriction enzymes for RdfA, Sph3, and volactin PCR products and BamHI/EcoRI restriction enzymes for the DdfA PCR product. This construction places the genes under control of the inducible tryptophanase promoter (p.*tna*). For complementation, deletion strains were transformed with the plasmid containing the corresponding gene or

with the empty vector pTA963 as a control (Supplementary Table 1). Δ*rdfA* complementation was performed using HS50 (Supplementary Table 1; Supplementary Fig. 2), as the initial deletion strain (HS37 in Supplementary Table 1) had retained part of the pTA131 plasmid.

## Live-cell imaging and analysis

*Hfx. volcanii* strains were inoculated from a single colony into 5 mL of Hv-Cab liquid medium with appropriate supplementation. Cultures were grown until they reached $OD_{600}$ values between 0.045 and 0.050 (early-log growth phase), 0.30 (mid-log growth phase), or $OD_{600}$ 1.15–1.80 (late-log growth phase). Three biological replicates were grown as described above, and for each culture, 1 mL aliquots were centrifuged at 4900 *g* for six minutes. Pellets were resuspended in their respective supernatant volumes: ~5 μL for early-log, ~50 μL for mid-log, and ~600 μL for late-log cultures. 1 μL of resuspension was placed on glass slides (Fisher Scientific) and a glass coverslip (Globe Scientific Inc.) was placed on top. Slides were visualized using a Leica Dmi8 inverted microscope attached to a Leica DFC9000 GT camera with Leica Application Suite X (version 3.6.0.20104) software, and both brightfield and differential interference contrast (DIC) images were captured at 100x magnification.

Phase contrast images of cells and fluorescence of volactin-msfGFP filaments were acquired under a Nikon TiE2 Widefield Inverted Microscope with a 100X CFI PlanApo Lambda 100x DM Ph3 objective. Cells were illuminated using the Lumencor Sola II Fluorescent LED at 488 nm using C-FL GFP HC HISN Zero Shift as emission filters. Images were recorded with a Hamamatsu ORCA Flash 4.0 v3 sCmos camera.

Brightfield images were quantified using CellProfiler (versions 4.2.1)[70]. Brightfield images were uploaded to CellProfiler and parameters were set for each set of images (strain, replicate, growth phase) based on specific image conditions to eliminate noise and maximize the number of identified cells. The CellProfiler pipeline used to analyze and quantify the images can be found at https://doi.org/10.5281/zenodo.8404691. Cell-specific data for each image set were exported, and aspect ratio (ratio of major to minor axes) was calculated. Statistical significance of aspect ratio comparisons between strains at each growth phase were assessed with an unpaired, nonparametric, two-tailed Kolmogorov-Smirnov test using GraphPad Prism version 9.3.1 and 10.0.2 for macOS (GraphPad Software, San Diego, California USA, www.graphpad.com). Effect size was calculated using PlotsOfDifferences[71].

Cell masks created from phase contrast images were generated through automated batch processing using Trackmate[72] running on Cellpose2[73]. A custom machine-trained model was used to segment rods and disks separately. From masks, cell aspect ratios were measured using Fiji. Subsequently, the fluorescence of volactin polymers was measured by first segmenting every structure with GFP signal using custom Fiji macros. Briefly, our custom macros generate volactin masks by first preprocessing images to subtract background and applying a Mexican Hat Filter to segment volactin puncta and filaments. Fluorescence measurements per cell were then made using standard Fiji plugins. The custom Fiji macros used here can be found at the Bisson Lab GitHub repository (https://github.com/Archaea-Lab/image-segmentation).

## Motility assays and motility halo quantification

Motility assays were assessed on 0.35% agar Hv-Cab medium with supplements as required. A toothpick was used to stab inoculate the agar followed by incubation at 45 °C. Motility assay plates were removed from the 45 °C incubator after two days and imaged after one day of room temperature incubation. Incubation at room temperature for one day allows darker pigmentation to be produced by the cells, making the halos easier to observe and image. Motility halos were quantified using Fiji (ImageJ2) (version 2.3.0/1.53q)[74]. Images were uploaded to Fiji, and the scale was set based on a 100 mm Petri dish

diameter. Statistical significance of halo diameters was assessed with an unpaired, two-tailed t-test using GraphPad Prism version 9.3.1 for macOS (GraphPad Software, San Diego, California USA, www.graphpad.com).

### Growth curve assay

Growth curves for wild type, Δ*rdfA*, Δ*ddfA*, Δ*sph3*, and Δ*volA\** strains were measured using a BioTek Epoch 2 microplate reader with BioTek Gen6 v.1.03.01 (Agilent). Colonies for each strain were inoculated in 5 mL liquid medium and grown at 45 °C with continuous shaking until they reached mid-log phase (OD$_{600}$ between 0.3 and 0.5). Cultures were diluted to an OD$_{600}$ of 0.01 with fresh liquid medium in a final volume of 200 μL in each well of a non-treated and flat-bottom polystyrene 96-well plate (Corning). Three biological replicates were used for each strain. 300 μL of media was aliquoted into each well for two rows of perimeter wells of the plate to help prevent evaporation of cultures. Readings were taken every 30 min with double orbital, continuous fast shaking (355 cpm, 4 mm) in between. Readings were taken at a wavelength of 600 nm for a total of approximately 96 h and then plotted in GraphPad Prism version 10.0.0 for macOS (GraphPad Software, San Diego, California USA, www.graphpad.com).

### Gene synteny analysis for *hvo_2174* and *hvo_2175*

A search for the eventual syntenic clustering of *rdfA* and *sph3* was performed against 120 haloarchaeal genomes with SyntTax (accessed Jan-2023)[75] (https://archaea.i2bc.paris-saclay.fr/SyntTax/). The positive result triggered a more extensive analysis. Using BLASTp (e-value cutoff 0.0001), an exhaustive set of homologs to RdfA was collected from 12 haloarchaeal species which are under continuous genome annotation survey[76,77]. Each of the identified sequences was in turn subjected to BLASTp analysis unless it was closely related to the query protein (>65% sequence identity). Each of these proteins was also compared by BLASTp to a set of genomes from 12 additional haloarchaeal genera.

A similar analysis was performed for Sph3. Additional criteria were used because of extensive similarity to other coiled-coil proteins (with BLASTp hits reaching e-values of e-30). For the set of genomes under continuous survey, annotation of BLASTp hits was taken into account. For genomes from other genera, BLASTp analysis was terminated once the first hit to an unrelated coiled-coil protein was encountered (e.g., repair ATPase Rad50).

All relevant homologs are listed in Supplementary Table 2. Proteins are listed in the same row if they are encoded adjacently (adjacent) or in genomic neighborhood (vicinity). When a homolog is reported for only one of the proteins, the absence of gene clustering is indicated by a dash. Adjacent genes were manually inspected to confirm the lack of the partner gene. Gene remnants (truncated pseudogenes) are not listed. The chromosome segregation protein SMC (e.g., HVO_0689) has orthologs in all analyzed genomes and is not listed in the table.

### Predicted start site for *ddfA*

dRNA-Seq and ribosome profiling data were previously obtained as described in[78,79]. These data were visualized with the Integrated Genome Browser[80] to identify the transcription start site of *ddfA*.

### Generation of chromosomal volactin-GFP fusion

A linear Gibson assembly[81] was created from four PCR fragments amplified using primers oBL174 and oBL175 (HVO_2015 upstream region, *Hfx. volcanii* genomic DNA as template), oBL354 and oBL24 (40aa-msfGFP, synthetic DNA as template), oHV6 and oHV7 (pyrE2 cassette, pTA962 as template), and oBL232 and oBL248 (HVO_2015 downstream region, *Hfx. volcanii* genomic DNA as template) (Supplementary Table 1). The assembly was then transformed into H26 cells as described above and selected on Hv-Cab plates without uracil. The

strain was confirmed by PCR, stocked in glycerol at −80 °C, and named aBL126 (Supplementary Table 1).

### Generation of chromosomal 2-color volactin-GFP / ftsZ1-mApple strain

To create the ftsZ1-mApple construct, a linear Gibson assembly[81] was created from four PCR fragments amplified using primers oHV3 and oHV4 (*ftsZ1* region, *Hfx. volcanii* genomic DNA as template), oHV140 and oHV170 (EG-mApple, synthetic DNA as template), oBL36 and oBL37 (mevR cassette, pMTCHA as template), and oBL308 and oHV171 (*ftsZ1* downstream region, *Hfx. volcanii* genomic DNA as template) (Supplementary Table 1). The assembly was then transformed into aBL126 cells as described above and selected on Hv-Cab plates without uracil. The strain was confirmed by PCR, stocked in glycerol at −80 °C, and then named aBL170 (Supplementary Table 1).

### 3-D super-resolution microscopy of volactin-msfGFP

aBL126 cells expressing volactin-msfGFP were grown as described above and membranes stained with 10 μg/μL FM5-95 (Invitrogen). A 2 μL droplet of culture was immediately transferred to 35 mm glass bottom dishes (Ibidi) and covered with a 1 × 1 cm of pre-warmed 0.25% agarose (Seakem). Cells were imaged at room temperature under a Nikon Ti-2 spinning disk W1-SoRa with 640 nm (cell membrane) and 488 nm (volactin-msfGFP) lasers and a Plan Apo λ 100x objective, Prime BSI express camera. Images were processed using NIS-Elements software (Nikon), using NIS.ai for denoising and 3-D deconvolution. Stacks were then rendered into 3-D images using Fiji.

### Plots for data visualization

All graphs for data visualization, unless stated otherwise, were plotted using PlotTwist[82] and PlotsOfData[83]. Cell shape images in Figs. 2a and b, 4e, and 6 were generated using Clip Studio Paint Pro, version 1.10.2.

### Reporting summary

Further information on research design is available in the Nature Portfolio Reporting Summary linked to this article.

## Data availability

The MS raw files generated for this study have been deposited to the ProteomeXchange Consortium (http://proteomecentral.proteomexchange.org) via the PRIDE partner repository[84] with the dataset identifier PXD040781, and all identification and quantification results have been included in the deposited dataset. Additional raw data for cell shape analyses and whole genome sequencing can be found on Zenodo (https://doi.org/10.5281/zenodo.8404691). Source data are provided with this paper, which includes cell shape quantification data for Figs. 3 and 5 along with Supplementary Figs. 1, 2, 4, and 8, growth curve data for Supplementary Fig. 3, motility quantification data for Fig. 3 along with Supplementary Fig. 8, and knockout strain gels for Supplementary Fig. 9. Additional data generated in this study that are provided in the Supplementary Information file include shape quantification data for Δ*hvo_B0194*, Δ*rdfA*, Δ*sph3*, Δ*ddfA*, and Δ*volA\** complementation strains, the growth curve for Δ*rdfA*, Δ*sph3*, Δ*ddfA*, and Δ*volA\**, transcriptional start site data for *ddfA*, clustering estimation and graphs, proteomics heat map, actin structural comparisons, Δ*volA\** motility data, and volactin fluorescence data. Source data are provided with this paper.

## Code availability

All scripts for the identification and quantification of peptides and proteins from MS datasets are made available through the Archaeal Proteome Project (http://archaealproteomeproject.org) and its associated GitHub repository (https://github.com/arcpp/ArcPP v1.4.0). The custom Fiji macros used in this study can be accessed through the

Bisson Lab GitHub repository (https://github.com/Archaea-Lab/image-segmentation v1.0.0).

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

## Acknowledgements

We thank Dr. Mike Dyall-Smith for help with analysis of genome sequences and Dr. Iain Duggin for providing the ΔcetZ1 strain. We gratefully acknowledge the quantitative proteomics analysis conducted by Dr. Jeremy L. Balsbaugh and Dr. Jennifer C. Liddle of the UConn Proteomics & Metabolomics Facility, a component of the Center for Open Research Resources and Equipment at the University of Connecticut. The authors would also like to acknowledge the NIH S10 High-End Instrumentation Award 1S10-OD028445-01A1, which supported this work by providing funds to acquire the Orbitrap Eclipse Tribrid mass spectrometer housed in the University of Connecticut Proteomics & Metabolomics Facility. We thank Dr. Simonetta Gribaldo and Dr. Nika Pende for insightful comments on our manuscript and Priyanka Chatterjee for providing the drawings of rod- and disk-shaped cells.

MP, AB, SS, HS, J Kouassi, YH, and AD were supported by the National Science Foundation Grant NSF-MBC2222076. HS was additionally supported by a National Institutes of Health Training Grant T32 AI 141393 "Microbial Pathogenesis and Genomics". AB is a Pew Scholar in the Biomedical Sciences, supported by The Pew Charitable Trusts. TR is supported by the Human Frontiers Science Program fund (RGY0074/2021), awarded to AB.

## Author contributions

H.S., D.S., F.P., A.B., S.S. and M.P. conceived the project and designed the experiments; H.S., Y.H., J. Kouassi, T.R., J. Kwak, A.D., D.S., A.M., F.P., A.B., S.S. and M.P. performed the experiments; H.S., Y.H., J. Kouassi, T.R., J. Kwak, A.D., D.S., A.M., F.P., A.B., S.S. and M.P. analyzed the data; H.S., A.B., S.S. and M.P. contributed materials and analysis tools; H.S., D.S., F.P., A.B., S.S. and M.P. wrote the paper. All authors reviewed the manuscript.

## Competing interests

The authors declare no competing interests.

## Additional information

**Peer review information** : *Nature Communications* thanks the anonymous reviewers for their contribution to the peer review of this work. A peer review file is available.

