## [Peer Review File NEW · Nature Communications]

Identification of structural and regulatory cell-shape determinants in *Haloferax volcanii*Editorial Note: This manuscript was initially submitted to, but not reviewed at, another journal. Mentions of the other journal have been redacted.

Reviewer #1 (Remarks to the Author):

Cell shape is an important characteristic of a given microbe and specific cell shapes often offer adaptation advantage for microbes. While we know a lot about bacterial cell shape regulation, relatively little is known about how archaeal cells maintain their shape and transition from one cell shape to another. *Haloferax volcanii* cells are rod-shaped in early-log growth phase but transition to flat, polygonal pleomorphic disk-shape in mid and late log growth phase. Rod shape is important for the motility of *H. volcanii* cells, while disk-shaped cells are defective in motility. Previous studies have found a few genes to be important for the cell shape of *H. volcanii*, such as *cetZ1*, which is a cytoskeletal protein important for rod-shaped morphology. However, apart from this, cell shape regulation of *H. volcanii* is largely unknown. In this study, Heather Schiller, et al. used a battery of approaches, including genetics, quantitative proteomics, and live-cell imaging, to identify genes important for cell shape of *H. volcanii* cells. They started with a clever genetic screen for hypermotile transposon mutants, followed by quantitative proteomics of mutants locked in rod-shaped and disk-shaped to identify shape specific genes. They successfully identified genes that are critical for rod-shape formation as well as genes whose disruption lock the cells in the disk-shaped. Strikingly, they identified an actin-like protein, named volactin, to be important for disk-shape morphology. They showed that volactin rapidly polymerizes and depolymerizes, and its level is correlated with the transition from rods to disks, thus uncovering an important function of archaeal actin-like proteins. Overall, this study has conducted a comprehensive study of cell shape determinants of *H. volcanii* and identified a number of genes important for cell shape.

Looking at the title of the manuscript, I was expecting the authors not only identified the genes important for cell shape but also did a detailed characterization of the genes, at least for a few of them, and revealed the mechanisms by which they regulate cell shape transition of *H. volcanii*. However, by the end, it was frustrating to see that the authors didn't dig into the working mechanisms of the genes they identified. The authors deserve compliments for the tremendous work to identify a whole new set of genes important for cell shape regulation, but it is a shame that they did not establish any connection among the newly identified genes and previously identified shape genes such as *cetZ1*. For examples, are *RdfA* and *Sph3* in the same pathway of *CetZ1* to regulate rod-shape formation? Do they interact with each other to control rod shape formation? Is there a connection between *DdfA* and volactin? Is there crosstalk between the genes critical for rod-shaped and disk-shaped formation? None of these questions have been addressed in the current manuscript. As a consequence, it is unclear how any of the genes affects cell shape. Thus, the manuscript appears to be prepared in a rush and not fully developed.

Major comments:

1. Page 5, line 98-103. The author isolated transposon mutant JK3 and JK5 which form rods only and used JK3 for subsequent proteomic experiments, however, they didn't even do a complementation experiment to confirm that the phenotype was due to the disruption of the *hvo_B0194* gene (also the *hvo_B0364*). In the main text, these two genes seem to be critical for disk-shape formation, in this sense, they should be identified in the quantitative proteomic comparison. However, it was not mentioned in the main text. It was hidden in the Supplemental Information that the shape defect of JK3 was not due to the disruption of *hvo_B0194* as a deletion strain of it had no shape defect. I think this information should not be hidden in the Supplemental Information, it should be explained in the main text, otherwise, readers would just wonder about it and question about the rationale of the design.
2. Page 7, line 147-151. The authors were unable to generate a *sph3* complete deletion strain, so that they used a partial deletion strain to characterize it. Have the authors tried to generate a depletion strain by replacing the promoter region of *sph3* with the tryptophan-inducible promoter *Ptna*? This seems to be a routine for characterization of essential genes in *H. volcanii* now. *H. volcanii* is well known for its polyploidy nature, it is not clear if the partial deletion mutant could be stably maintained. A depletion strain should be a much reliable mutant to characterize the gene. Or at least a complementation test should have been included to validate the function of *sph3*.
3. Page 7, line 152-159 and related Figure 3. Here the authors showed the deletion of *hvo_2174* resulted in the formation of disk cells, but a complementation test was not done to confirm the phenotype. The defect was highly likely due to deletion of *hvo_2174*, but a complementation test is still necessary to rule out other possibilities, such as polar effect on the neighboring genes. A complementation test for the *hvo_2176* deletion mutant was provided, not sure why the authors

didn't do a complementation test for hvo_2174.

4. Page 10, line 233-244. Similar to the situation of sph3, the authors used a partial deletion mutant of volA to characterize it. A depletion strain should be more reliable and complementation test should have been included.

5. Page 35, line 764-769, Figure 5E, here the authors were looking at the volactin fluorescent protein fusion in early and mid log phase, it is not necessary to use two different colors for the same protein fusion. Two different colors just gave readers a false impression that they represent two different proteins.

6. The discussion section is over simplified and a lot of information has been allocated to the Supplemental Information. In my opinion, many of the points in the supplemental information should have been placed in the discussion section in the main text, such as the discussion on volactin.

7. As pointed out above, a major caveat of the manuscript is the lack of connection among the identified genes, how they collaborate to regulate cell shape in *H. volcanii*. Some simple experiments could potentially establish a relationship among the cell shape determinants and strengthen the story. For example, subcellular localization of RdfA, DdfA, Sph3 and whether their absence affects the localization of the CetZ1 and vice versa.

Minor comments:

1. Page 2, line 28, abstract, add "putative" between "including" and "transporters". These genes were annotated as transporters, transducers and so on, they have not been characterized.

2. Page 11, line 261, "actin-binding factors" should be "volactin-binding factors".

3. Page 12, line 295-296. It was claimed here that "many transducers appeared to be important for rods." however, it was not confirmed in the study that many of the genes annotated as transducers are important for rods, please rephrase.

4. Page 13, line 309-310. There is no evidence that DdfA regulates cell shape through interactions with RdfA and Sph3. There is no experiments presented in the study that tried to establish a relationship among DdfA, RdfA, Sph3, CetZ1 and Volactin.

5. Page 38, line 789, sph3 and volA should be italic.

Reviewer #2 (Remarks to the Author):

In this manuscript, the authors describe how they use cell-shape mutants of the euryarchaeon *Haloferax volcanii* to identify other proteins involved in determining cell morphology by mass spectrometry. They found several interesting protein candidates that likely play a role at different stages at the transitioning of *Haloferax* cells from rod-shaped in the early log phase to disk shape in the late log phase. The authors show that RdfA, Sph3, and DdfA play important roles in this process by constructing deletion mutants and testing these for their shape throughout the growth curve and on motility plates. Moreover, they identify Volactin, an actin homolog, which forms filaments predominantly in disk shaped cells, and seems to be important for the transition from rod to disk shaped cells.

Do the RdfA, Sph3 and DdfA interact at some point? Did the authors check where these proteins are localized in the cell? Do they interact at any point of the cell development with the volactin?

Please add something like an operon map or the cluster you are working with, do the genes overlap, is anything known about their transcriptional start sites etc...

Was the delta hvo_2176 constructed within this paper or is this the one in SAH1 described earlier?

The authors show complementation of the Hvo_2176 deletion mutant, but I could not find complementation assays for the Hvo_2175 and Hvo_2174 deletion mutants.

Hvo_2174 and volactin could not be fully deleted as a few copies remained; how stable is this genomic situation? If the genes are so important, does the copy number of the genome copies containing the essential genes not increase in time? In other *H. volcanii* studies this has been solved by replacing the native promoter of essential genes by a tryptophane inducible promoter

to be able to check the effect of the full depletion of the gene. Why did the authors not take this approach?

Did the authors consider deleting the predicted regulators, like TrmB1. To analyze whether these indeed have an effect on the regulatory network concerning the shape change?

Extended Data Figure 1: please add a map where in the gene the different primer pairs bind.

Extended data figure 2: what is the strange kink in all the growth curves shown after around 25 hr? Was the shaker just standing still? But then no technical replicates were done? Or is this behavior repeatable?

Extended Data Table 1: why not show the visual output of Syntax here? Much easier to grasp and more intuitive? What about Hvo_2176, is it not included in the cluster?

Line 175: the description of the experiment is now very basic in the manuscript. Please describe that you sampled the cells at different growth phase stages.

Motility assays: two days at 45C and 1 day at room temperature? Why is the one day at room temperature step required?

Fig 5E: in "early log" Volactin seems pinkish and in "mid log" blueish, please choose the same colour. The scale bar in "mid log" is very large compared to the image, please reduce the size. This is true for all images shown in Figure 5, the sizes of the scale bar vary a lot in, A it is black without number, please change it to make it a little bit more coherent.

Line 394 Supplemental discussion: remove "inner"

Reviewer #3 (Remarks to the Author):

- What are the noteworthy results?

The authors identified several new genes participating in cell-shape determination of *H. volcanii*. For their identification, they leveraged a clever and powerful approach of combining phenotypic screening, forward and reverse genetics, and proteomics.

- Will the work be of significance to the field and related fields? How does it compare to the established literature? If the work is not original, please provide relevant references.

In light of the scarce existing knowledge about archaeal cell shape establishment and modulation, this work is a significant contribution to this field and relevant for the wider field of microbiology by giving further insight whether involved genes and their regulations are conserved in and across microbial kingdoms.

The work builds upon previous knowledge about the morphology and motility of *H. volcanii* that was key to successful screening. However, the overall experimental approach and the results display a high level of novelty.

- Does the work support the conclusions and claims, or is additional evidence needed?

In my opinion, the only case where more evidence would be helpful is the presence of *ddfA* in WT. Without a doubt, the genetic experiments and microscopy demonstrate its function in cell shape determination. Yet, the authors failed to identify *ddfA* when comparing the proteomes of WT and mutant and did not provide alternative data that confirms its presence.

- Are there any flaws in the data analysis, interpretation and conclusions? Do these prohibit publication or require revision?

I feel only confident to judge the proteomics data, where I requested revisions to facilitate data sharing and re-use.

- Is the methodology sound? Does the work meet the expected standards in your field?

Methodology is sound and at least for the proteomics done I can say it meets the expected standards in the field.

- Is there enough detail provided in the methods for the work to be reproduced?

All methods related to proteomics provide sufficient detail, the others I cannot judge.

- Other, own points

Please briefly discuss how likely it is that other cell shape-determining genes have been missed by the screening strategy for hypermotility? Are there any other cell shapes observed for *H. volcanii* that were not targeted in this work?

In case the manuscript becomes accepted, please upload the proteome data and results in a public format to ProteomXchange. I recommend "complete submission" that enables viewing of data without vendor tools.

Figure 7B: Do the proteins bind ATP and if yes, should there not be a binding site for phosphate 3?

Extended Data Table 1: How do you explain fact that some organisms only contain one of the two homologs? How would this affect your model for cell shape determination (Figure 6) in these organisms?

Extended Data Table 3: Instruments were set to exclude charge states 1, >6. How did you handle unknown charge states? For the AGC of Eclipse, I suggest to replace % with the more meaningful number of ions – the software allows you to determine number of ions for given %.

Reviewer #1 (Remarks to the Author):

Cell shape is an important characteristic of a given microbe and specific cell shapes often offer adaptation advantage for microbes. While we know a lot about bacterial cell shape regulation, relatively little is known about how archaeal cells maintain their shape and transition from one cell shape to another. *Haloferax volcanii* cells are rod-shaped in early-log growth phase but transition to flat, polygonal pleomorphic disk-shape in mid and late log growth phase. Rod shape is important for the motility of *H. volcanii* cells, while disk-shaped cells are defective in motility. Previous studies have found a few genes to be important for the cell shape of *H. volcanii*, such as *cetZ1*, which is a cytoskeletal protein important for rod-shaped morphology. However, apart from this, cell shape regulation of *H. volcanii* is largely unknown. In this study, Heather Schiller, et al. used a battery of approaches, including genetics, quantitative proteomics, and live-cell imaging, to identify genes important for cell shape of *H. volcanii* cells. They started with a clever genetic screen for hypermotile transposon mutants, followed by quantitative proteomics of mutants locked in rod-shaped and disk-shaped to identify shape specific genes. They successfully identified genes that are critical for rod-shape formation as well as genes whose disruption lock the cells in the disk-shaped. Strikingly, they identified an actin-like protein, named volactin, to be important for disk-shape morphology. They showed that volactin rapidly polymerizes and depolymerizes, and its level is correlated with the transition from rods to disks, thus uncovering an important function of archaeal actin-like proteins. Overall, this study has conducted a comprehensive study of cell shape determinants of *H. volcanii* and identified a number of genes important for cell shape.

We would like to thank the reviewer for their supportive comments, and we appreciate the excellent summary of our manuscript. In addition, we would like to stress one aspect more clearly: actin homologs have only rarely been identified in archaea, especially in the euryarchaeota, and none of those identified has yet been experimentally characterized. Thus, we not only report the identification of a previously undetected euryarchaeal actin homolog but also report its first experimental characterization. We determined that this protein shows actin-like physiological behavior, such as polymerization and depolymerization of filaments, and we carried out advanced 3D imaging to determine the distribution of the volactin. We have stressed our achievements more clearly in the abstract (line 35-36).

Looking at the title of the manuscript, I was expecting the authors not only identified the genes important for cell shape but also did a detailed characterization of the genes, at least for a few of them, and revealed the mechanisms by which they regulate cell shape transition of *H. volcanii*. However, by the end, it was frustrating to see that the authors didn't dig into the working mechanisms of the genes they identified. The authors deserve compliments for the tremendous work to identify a whole new set of genes important for cell shape regulation, but it is a shame that they did not establish any connection among the newly identified genes and previously identified shape genes such as *cetZ1*. For examples, are *RdfA* and *Sph3* in the same pathway of *CetZ1* to regulate rod-shape formation? Do they interact with each other to control rod shape formation? Is there a connection between *DdfA* and volactin? Is there crosstalk between the genes critical for rod-shaped and disk-shaped formation? None of these questions have been addressed in the current manuscript. As a consequence, it is unclear how any of the genes affects cell shape. Thus, the manuscript appears to be prepared in a rush and not fully developed.

We would like to thank the reviewer for recognizing the “battery of approaches” and “tremendous work to identify a whole new set of genes important for cell shape regulation”, which resulted in a “comprehensive study of cell shape determinants of *Hfx. volcanii* and identified a number of genes important for cell shape”. We also want to highlight that we indeed provided extensive characterizations of several of the newly identified shape determinants, including extensive 3D imaging, which confirms that the newly identified actin homolog polymerizes into filaments. Additionally, for other components we provided experimental data through a series of deletion (or partial deletion) mutants and cell biological assays, which confirmed their involvement in shape determination (together with data on motility), thus validating multiple candidates that were identified through the combination of genomic and proteomic analyses. These characterizations include several proteins that were previously designated as proteins of unknown function, i.e., proteins for which not even similar functional domains were known. Since the last submission we were also able to obtain a complete knockout of *sph3* and carried out complementation experiments for the Δhvo_2174 , $\Delta sph3$ (lines 160-165 and Supplementary Fig. 2) and $\Delta volA$ strains (lines 313-314 and Supplementary Fig. 8d), all supporting that these proteins are novel components involved in cell shape determination. We are as interested as the reviewer in learning more about the details and mechanisms underlying the observed phenotypes. However, the complexity of the scientific questions at hand makes these questions far beyond the scope of the current manuscript. Despite these characterizations, we have removed ‘characterization’ from the title to deemphasize this point (line 1).

Major comments:

1. Page 5, line 98-103. The author isolated transposon mutant JK3 and JK5 which form rods only and used JK3 for subsequent proteomic experiments, however, they didn’t even do an complementation experiment to confirm that the phenotype was due to the disruption of the *hvo_B0194* gene (also the *hvo_B0364*). In the main text, these two genes seem to be critical for disk-shape formation, in this sense, they should be identified in the quantitative proteomic comparison. However, it was not mentioned in the main text. It was hidden in the Supplemental Information that the shape defect of JK3 was not due to the disruption of *hvo_B0194* as a deletion strain of it had no shape defect. I think this information should not be hidden in the Supplemental Information, it should be explained in the main text, otherwise, readers would just wonder about it and question about the rationale of the design.

We agree with the reviewer and have moved the shape phenotype of the Δhvo_B0194 strain from the Supplementary Discussion into the main text (lines 145-149). We also included the microscopy images and quantification of the shape of this strain as a supplementary figure (Supplementary Fig. 1). However, we want to emphasize that despite the fact that the gene affected by the transposon in JK3 is unrelated to the shape defect, the rod-only phenotype of JK3 was still highly relevant in the identification of shape determinants via quantitative proteomics.

2. Page 7, line 147-151. The authors were unable to generate a *sph3* complete deletion strain, so that they used a partial deletion strain to characterize it. Have the authors tried to generate a depletion strain by replacing the promoter region of *sph3* with the tryptophan-inducible promoter *Ptna*? This seems to be a routine for characterization of essential genes in *H. volcanii* now. *H. volcanii* is well known for its polyploidy nature, it is not clear if the partial deletion mutant could be stably maintained. A depletion

strain should be a much reliable mutant to characterize the gene. Or at least a complementation test should have been included to validate the function of *sph3*.

As noted above, additional attempts to make a complete *sph3* knockout were successful. We also confirmed that the Δ *sph3* morphology and motility phenotype is due to the lack of Sph3 by complementing the strain with Sph3 expressed from a plasmid (lines 162-165 and Supplementary Fig. 2).

3. Page 7, line 152-159 and related Figure 3. Here the authors showed the deletion of *hvo_2174* resulted in the formation of disk cells, but a complementation test was not done to confirm the phenotype. The defect was highly likely due to deletion of *hvo_2174*, but a complementation test is still necessary to rule out other possibilities, such as polar effect on the neighboring genes. A complementation test for the *hvo_2176* deletion mutant was provided, not sure why the authors didn't do a complementation test for *hvo_2174*.

We performed gene complementation for Δ *rdfA* via an inducible plasmid and confirmed that the Δ *rdfA* morphology and motility phenotype is due to the lack of RdfA by complementing the strain with *rdfA* expressed from a plasmid (lines 162-165 and Supplementary Fig. 2)

4. Page 10, line 233-244. Similar to the situation of *sph3*, the authors used a partial deletion mutant of *volA* to characterize it. A depletion strain should be more reliable and complementation test should have been included.

Unlike with *sph3*, additional attempts to make a *volA* deletion strain were unsuccessful, supporting that this gene is essential. However, we confirmed that the Δ *volA* disk-formation and motility phenotypes are due to the lack of VolA* by complementing the strain with Vol* expressed from a plasmid (lines 313-314 and Supplementary Fig. 8d).

It should be noted that the "partial deletion strain" is actually one variant of a "depletion strain", and the latter would most likely result in similar phenotypes as the former, especially given the common issue of promoter leakiness. Furthermore, if *volA* is in fact essential, cells may die and preclude unambiguous phenotypic observations. While the currently available Δ *volA** strain may not be optimal to glean more detailed insights into the molecular mechanisms, the obtained cell biological data are valid and show a clear phenotype that, upon complementation in trans as noted above, confirm involvement of *volA* in cell-shape determination. Overall, the additional information gained from these experiments would be very limited, as the partial deletion of *volA* already strongly supports the involvement of *volA* in disk formation.

5. Page 35, line 764-769, Figure 5E, here the authors were looking at the volactin fluorescent protein fusion in early and mid log phase, it is not necessary to use two different colors for the same protein fusion. Two different colors just gave readers a false impression that they represent two different proteins.

We amended the colors so the same one is used for each growth phase (Fig. 5).

6. The discussion section is over simplified and a lot of information has been allocated to the Supplemental Information. In my opinion, many of the points in the supplemental information should have been placed in the discussion section in the main text, such as the discussion on volactin.

We have expanded the discussion. The oversimplified discussion was due to the stricter word count (3,000 words) for our initial submission to [Redacted] (green highlights: lines 134-141, 145-149, 169-176, 185-197, 209-212, 226-230, 254-285, 411-426 transferred from original supplemental discussion).

7. As pointed out above, a major caveat of the manuscript is the lack of connection among the identified genes, how they collaborate to regulate cell shape in *H. volcanii*. Some simple experiments could potentially establish a relationship among the cell shape determinants and strengthen the story. For example, subcellular localization of RdfA, DdfA, Sph3 and whether their absence affects the localization of the CetZ1 and vice versa.

Building on the results obtained by us and reported in the current manuscript, experiments of the kind indicated by the reviewer are valid and highly relevant options for future analyses. In fact, we have performed localization studies for volactin and confirmed its rapid polymerization and depolymerization through live cell imaging. However, performing similar experiments (and/or co-localization studies with CetZ1) with RdfA, DdfA, and Sph3 is beyond the scope of the current manuscript. The number of proteins that we identified to be involved in cell-shape determination through a combination of genomics, proteomics, and cell biology, as well as the large number of interesting follow-up experiments that have been suggested, highlight the novelty and power of this approach to provide new insights into cell-shape determination in archaea. Thereby, our results set the stage for numerous co-localization studies and detailed biochemical characterizations that can be performed in future analyses.

Minor comments:

1. Page 2, line 28, abstract, add “putative” between “including” and “transporters”. These genes were annotated as transporters, transducers and so on, they have not been characterized.

We have amended this in the text but have elected to use ‘predicted’ rather than ‘putative’ (line 29).

2. Page 11, line 261, “actin-binding factors” should be “volactin-binding factors”.

We have amended this in the text (line 335).

3. Page 12, line 295-296. It was claimed here that “many transducers appeared to be important for rods.” however, it was not confirmed in the study that many of the genes annotated as transducers are important for rods, please rephrase.

We have rephrased this sentence to imply association rather than ‘importance for’ rod formation (line 370).

4. Page 13, line 309-310. There is no evidence that DdfA regulates cell shape through interactions with RdfA and Sph3. There is no experiments presented in the study that tried to establish a relationship among DdfA, RdfA, Sph3, CetZ1 and Volactin.

We have amended this sentence to instead say that it is unresolved whether DdfA interacts with RdfA and Sph3 (lines 383-384)

5. Page 38, line 789, sph3 and volA should be Italic.

Our figure titles are in italics, necessitating that any text that is normally in italics instead be non-italicized.

Reviewer #2 (Remarks to the Author):

In this manuscript, the authors describe how they use cell-shape mutants of the euryarchaeon *Haloferax volcanii* to identify other proteins involved in determining cell morphology by mass spectrometry. They found several interesting protein candidates that likely play a role at different stages at the transitioning of *Haloferax* cells from rod-shaped in the early log phase to disk shape in the late log phase. The authors show that RdfA, Sph3, and DdfA play important roles in this process by constructing deletion mutants and testing these for their shape throughout the growth curve and on motility plates. Moreover, they identify Volactin, an actin homolog, which forms filaments predominantly in disk shaped cells, and seems to be important for the transition from rod to disk shaped cells.

Do the RdfA, Sph3 and DdfA interact at some point? Did the authors check where these proteins are localized in the cell? Do they interact at any point of the cell development with the volactin?

We would like to thank the reviewer for their supportive comments. We agree that the questions outlined are interesting and worth investigating further. However, considering the intensive analysis already provided, together with the number of proteins identified to be involved in cell-shape determination, we believe that these experiments are beyond the scope of the manuscript. Please also see our response to Reviewer #1 for further details.

Please add something like an operon map or the cluster you are working with, do the genes overlap, is anything known about their transcriptional start sites etc...

We have added this information to the manuscript (Supplementary table 1 and lines 198 to 200 in main text, gene map in Fig. 3f)

Was the Δhvo_2176 constructed within this paper or is this the one in SAH1 described earlier?

Yes, Δhvo_2176 ($\Delta ddfA$) was constructed within this paper. SAH1 is a transposon insertion mutant, and the assignment of the position of the insertion was based on the previous annotation of *ddfA* and thus was implicated to be upstream of *ddfA*. We have clarified this point in the results section of the manuscript (lines 181-184).

The authors show complementation of the Hvo_2176 deletion mutant, but I could not find complementation assays for the Hvo_2175 and Hvo_2174 deletion mutants.

We performed complementation assays for $\Delta rdfA$ and $\Delta sph3$ and included these data in the resubmitted manuscript (Supplemental Figure 2 Lines 162-165).

Hvo_2174 and volactin could not be fully deleted as a few copies remained; how stable is this genomic situation? If the genes are so important, does the copy number of the genome copies containing the essential genes not increase in time? In other *H. volcanii* studies this has been solved by replacing the native promoter of essential genes by a tryptophane inducible promoter to be able to check the effect of the full depletion of the gene. Why did the authors not take this approach?

As noted above we now have a complete knockout of *sph3*. Moreover, please see our reply to Reviewer #1 about the use of the partial *volA* deletion strain and the limited additional information that would be gained from the use of tryptophan-inducible promoter.

Did the authors consider deleting the predicted regulators, like TrmB1. To analyze whether these indeed have an effect on the regulatory network concerning the shape change?

While we agree that assessing the specific role of regulators in shape change is interesting, the generation and characterization of a *trmB1*-deletion strain is beyond the scope of the paper.

Extended Data Figure 1: please add a map where in the gene the different primer pairs bind.

We have added a primer binding map to this figure (Supplementary Figure 9b).

Extended data figure 2: what is the strange kink in all the growth curves shown after around 25 hr? Was the shaker just standing still? But then no technical replicates were done? Or is this behavior repeatable?

We repeated the growth curve experiment and included the new graph (Supplementary Fig. 3).

Extended Data Table 1: why not show the visual output of Syntax here? Much easier to grasp and more intuitive? What about Hvo_2176, is it not included in the cluster?

Supplementary Table 1 shows that RdfA and Sph3 are typically clustered. DdfA (HVO_2176) is clustered in *Hfx. volcanii* but rarely clustered in other genomes. We have added this clarification to the text (lines 198-200). For RdfA/Sph3, there is no simple Syntax output for the analysis results. (i) Syntax does not show plasmid-encoded proteins, several of which are on the list. (ii) Syntax shows only the best homolog, while many listed proteins are paralogous and thus would not be displayed. (iii) RdfA proteins are only distantly related, and these distant homologs might escape detection by Syntax. (iv) Sph3 proteins show significant similarity to unrelated coiled-coil proteins, and Syntax may not be sufficiently discriminative to avoid the assignment of the same color.

Line 175: the description of the experiment is now very basic in the manuscript. Please describe that you sampled the cells at different growth phase stages.

We have expanded our description of this method (lines 206-207).

Motility assays: two days at 45C and 1 day at room temperature? Why is the one day at room temperature step required?

After two days at 45°C, the cells swim outward and form the motility halo. Incubation at room temperature for one day allows darker pigmentation to be produced by the cells, and thus the halos are easier to observe and image. We now state this clarification in the text (lines 668-669).

Fig 5E: in “early log” Volactin seems pinkish and in” mid log” blueish, please choose the same colour. The scale bar in “mid log” is very large compared to the image, please reduce the size. This is true for all images shown in Figure 5, the sizes of the scale bar vary a lot in, A it is black without number, please change it to make it a little bit more coherent.

We have amended the figure to use the same colors in the early- and mid-log growth phase images. We have adjusted the size of the scale bars to match, and we will use either black or white scale bars based on the background of the image.

Line 394 Supplemental discussion: remove “inner”

We have deleted this word from the sentence.

Reviewer #3 (Remarks to the Author):

- What are the noteworthy results?

The authors identified several new genes participating in cell-shape determination of *H. volcanii*. For their identification, they leveraged a clever and powerful approach of combining phenotypic screening, forward and reverse genetics, and proteomics.

- Will the work be of significance to the field and related fields? How does it compare to the established literature? If the work is not original, please provide relevant references.

In light of the scarce existing knowledge about archaeal cell shape establishment and modulation, this work is a significant contribution to this field and relevant for the wider field of microbiology by giving further insight whether involved genes and their regulations are conserved in and across microbial kingdoms.

The work builds upon previous knowledge about the morphology and motility of *H. volcanii* that was key to successful screening. However, the overall experimental approach and the results display a high level of novelty.

- Does the work support the conclusions and claims, or is additional evidence needed?

In my opinion, the only case where more evidence would be helpful is the presence of *ddfA* in WT. Without a doubt, the genetic experiments and microscopy demonstrate its function in cell shape determination. Yet, the authors failed to identify *ddfA* when comparing the proteomes of WT and mutant and did not provide alternative data that confirms its presence.

We would like to thank the reviewer for their supportive comments that recognized “the clever and powerful approach” and “high level of novelty” of combining proteomic and genetic analyses. Detecting *DdfA* using bottom-up proteomics is complicated by several factors: (i) the small size of the protein results in few theoretically detectable peptides, and within the datasets of the Archaeal Proteome Project, it could only be found if the sample preparation did not include the use of filter units; (ii) the atypical (yet unresolved) start codon means that the full native sequence of the protein was not included in the protein database search. However, by inserting a typical start codon upstream of the transcribed region, we were able to complement the mutant phenotype. This complementation represents an alternative approach to the proteomics analyses, and in our view, at least partially confirmed the existence and biological relevance of *DdfA*.

- Are there any flaws in the data analysis, interpretation and conclusions? Do these prohibit publication or require revision?

I feel only confident to judge the proteomics data, where I requested revisions to facilitate data sharing and re-use.

- Is the methodology sound? Does the work meet the expected standards in your field?

Methodology is sound and at least for the proteomics done I can say it meets the expected standards in the field.

- Is there enough detail provided in the methods for the work to be reproduced?

All methods related to proteomics provide sufficient detail, the others I cannot judge.

- Other, own points

Please briefly discuss how likely it is that other cell shape-determining genes have been missed by the screening strategy for hypermotility? Are there any other cell shapes observed for *H. volcanii* that were not targeted in this work?

In lines 112-114 we write “...inherent limitations of genetic screens (Supplementary Discussion 1) prevent their use for more comprehensive analyses of shape-transition pathways”, and these limitations are described in detail in Supplementary Discussion 1. Rods and disks are the only shapes that have been observed for *Hfx. volcanii* under the conditions tested.

In case the manuscript becomes accepted, please upload the proteome data and results in a public format to ProteomXchange. I recommend “complete submission” that enables viewing of data without vendor tools.

We have uploaded the dataset to PRIDE. We have included the project name and accession number along with reviewer login information below:

Project Name: Comparative, quantitative proteomic analysis of cell shape transitions in *Haloferax volcanii*

Project accession: PXD040781

Reviewer account username: reviewer_pxd040781@ebi.ac.uk

Reviewer account password: CHc6tghX

A "complete submission" is not possible due to the data format restrictions imposed by ProteomeXchange, but the results will be made available through the Archaeal Proteome Project at the point of publication (and can be either downloaded in full from the corresponding Zenodo repository or viewed through the interactive Archaeal Proteome Project website).

Figure 7B: Do the proteins bind ATP and if yes, should there not be a binding site for phosphate 3?

Based on the predicted structure we provide for volactin, there is clear indication for ATP binding. However, we did not include this in the manuscript because the structure is predicted and not resolved.

Extended Data Table 1: How do you explain fact that some organisms only contain one of the two homologs? How would this affect your model for cell shape determination (Figure 6) in these organisms?

Most species encode at least one gene pair but may encode additional singlets. Two species (*Halopiger xanaduensis*, *Natronobacterium gregoryi*) do not encode a gene pair but encode both proteins, which thus still might cooperate. There is no species that codes for only one of the proteins and not for the other.

Extended Data Table 3: Instruments were set to exclude charge states 1, >6. How did you handle unknown charge states? For the AGC of Eclipse, I suggest to replace % with the more meaningful number of ions – the software allows you to determine number of ions for given %.

We have amended the Supplementary table 3 as follows: we have added 'unknown' to the excluded charge states, and for the MS1 and MS2 AGC targets, we have added 250% (1e6) and 200% (1e5), respectively.

Reviewer #1 (Remarks to the Author):

The authors have addressed my concerns and other reviewer's comments well. The revised manuscript has been substantially improved. Although I still think addition of some localization studies of the identified shape determinants will further strengthen the paper, the authors' argument for not doing this kind of experiments in this study is reasonable. As I commented for the first submission, the identification of a whole new set of genes important for archaeal cell shape transition is of sufficient novelty and significance. My suggestions were meant to improve and strengthen the paper. If the manuscript is eventually accepted, I suggest the authors to place some of the supplementary figures as main figures as the journal could have 10 displayed items. In this way, the main article might read better. However, this is up to the authors and the editorial team.

Minor comments:

1. Supplementary figures 2, 3, 4 and 8 could be reallocated into the main article.
2. Line 152-153, add reference or databases where they are annotated.
3. Line 306, "..., we attempted to generate Δ volA." Change to "..., we attempted to generate a Δ volA strain and characterize it."
4. Line 307-308. Not all readers know *Hfx volcanii* cells are polyploid, so it is better to clearly point this out here or somewhere else in the paper so that readers can understand what a "partial deletion" really means.
5. Line 321-322. Does the strain with volA-msfGFP behave similarly to wild type? I assume there is no difference, it should be clearly pointed out.

Reviewer #2 (Remarks to the Author):

The authors have constructed a new sph3 mutant and characterized it. Furthermore, they performed complementation assays for the presented mutants and could show that the wild type phenotype was restored in the mutants.

Probably during these experiments, it seems that the authors realized that the originally used Δ rdfA strain (HS37) still had a plasmid integrated in the genome which was used for the deletion of the gene. Now, they created a new clean deletion mutant for the complementation assays (HS50).

Therefore, it seems that the authors have used a Δ rdfA strain with an integrated plasmid for the proteomic analyses. This should now be clearly stated in the manuscript in the results part. It is known that the presence of plasmids influences the timing of when *H. volcanii* cells change their cell shape which is important for this particular experiment.

Reviewer #3 (Remarks to the Author):

The authors adequately addressed all my critics and questions.

Point-by-point responses to the reviewers' comments:

Reviewer 1

The authors have addressed my concerns and other reviewer's comments well. The revised manuscript has been substantially improved. Although I still think addition of some localization studies of the identified shape determinants will further strengthen the paper, the authors' argument for not doing this kind of experiments in this study is reasonable. As I commented for the first submission, the identification of a whole new set of genes important for archaeal cell shape transition is of sufficient novelty and significance. My suggestions were meant to improve and strengthen the paper. If the manuscript is eventually accepted, I suggest the authors to place some of the supplementary figures as main figures as the journal could have 10 displayed items. In this way, the main article might read better. However, this is up to the authors and the editorial team.

Supplementary figures 2, 3, 4 and 8 could be reallocated into the main article.

We have decided to keep these figures as supplementary figures.

2. Line 152-153, add reference or databases where they are annotated.

The sentence now reads: "NCBI records indicate that HVO_2174 is a protein of unknown function, and HVO_2175 is annotated as Sph3..."

3. Line 306, "... we attempted to generate Δ volA." Change to "... we attempted to generate a Δ volA strain and characterize it."

The sentence now reads "... we attempted to generate a Δ volA strain."

4. Line 307-308. Not all readers know *Hfx volcanii* cells are polyploid, so it is better to clearly point this out here or somewhere else in the paper so that readers can understand what a "partial deletion" really means.

The sentence now reads: "... we could only isolate a partial deletion of this polyploid haloarchaeon, which retained at least one gene copy..."

5. Line 321-322. Does the strain with volA-msfGFP behave similarly to wild type? I assume there is no difference, it should be clearly pointed out.

Yes, the volA-msfGFP strain behaves similarly to wild type. We have included a new graph in Supplementary Figure 8 to show this.

Reviewer #2

The authors have constructed a new sph3 mutant and characterized it. Furthermore, they performed complementation assays for the presented mutants and could show that the wild type phenotype was restored in the mutants.

Probably during these experiments, it seems that the authors realized that the originally used Δ rdfA strain (HS37) still had a plasmid integrated in the genome which was used for the deletion of the gene. Now, they created a new clean deletion mutant for the complementation assays (HS50).

Therefore, it seems that the authors have used a Δ rdfA strain with an integrated plasmid for the

proteomic analyses. This should now be clearly stated in the manuscript in the results part. It is known that the presence of plasmids influences the timing of when *H. volcanii* cells change their cell shape which is important for this particular experiment.

We now added the following sentences: “It should be noted that the initial *rdfA* deletion strain (HS37) used for the proteomics, had retained part of the pTA131 plasmid. However, the cell shape phenotypes of this Δ *rdfA* strain (Fig. 3) and the subsequent deletion strain of *rdfA* (HS50) devoid of pTA131, used for the complementation experiments (Supp. Fig. 2), are the same.” (lines 298-301).